# United Minds or Isolated Agents? Exploring Coordination of LLMs under Cognitive Load Theory

**HaoYang Shang**[*]                                                   *info.breathingcore@gmail.com*
*Hong Kong University of Science and Technology*

**Xuan Liu**[*]                                                              *xul049@ucsd.edu*
*University of California San Diego*

**Zi Liang**                                                *zi1415926.liang@connect.polyu.hk*
*Hong Kong Polytechnic University*

**Jie Zhang**                                                            *csejzhang@ust.hk*
*Hong Kong University of Science and Technology*

**Haibo Hu**                                                            *haibo.hu@polyu.edu.hk*
*Hong Kong Polytechnic University*

**Song Guo**                                                          *songguo@cse.ust.hk*
*Hong Kong University of Science and Technology*

**Reviewed on OpenReview:** *https://openreview.net/forum?id=8BYJHZiZ5T*

## Abstract

Large Language Models (LLMs) exhibit a notable performance ceiling on complex, multi-faceted tasks. As practitioners increasingly rely on heavy *context engineering*—curating intricate instructions, tool schemas, and multi-turn histories—the processing demands often exceed the LLM's effective attention budget, leading to *context rot*. Drawing an analogy to Cognitive Load Theory (CLT) in cognitive science, we propose that this bottleneck is functionally analogous to the bounded working memory of the human mind. Rather than relying on heuristic prompt engineering, we use CLT as a principled *design lens* for LLM system design. To operationalize this insight, we introduce ***CoThinker***, an instantiation of a CLT-driven multi-agent framework. *CoThinker* operationalizes CLT principles by distributing intrinsic cognitive load through agent specialization and managing transactional load via structured communication and a collective working memory. We empirically evaluate *CoThinker* on complex problem-solving tasks and fabricated high cognitive load scenarios. Our results are consistent with a CLT-informed account of multi-agent coordination: gains concentrate on reasoning-heavy tasks where cognitive load is high, while coordination overhead dominates on low-intrinsic-load tasks such as instruction-following—a boundary predicted by the cognitive-load-profile view. Our analysis reveals characteristic interaction patterns that cast insights from collective cognition and load management into a principled approach to agent system design.

## 1 Introduction

As Large Language Models (LLMs) are deployed on increasingly complex real-world tasks (Chang et al., 2024; Zhao et al., 2024; Li et al., 2024), practitioners increasingly rely on *context engineering*—the careful

---

[*]Equal contribution.

curation of information presented to a model so that its finite attention budget is spent effectively (Rajasekaran et al., 2025). To steer LLM behavior for complex agentic workflows, developers aggressively pack the context window with intricate system instructions, few-shot demonstrations, exhaustive tool specifications, and expanding multi-turn histories (Wang et al., 2024a; Guo et al., 2024). While this context-heavy paradigm shapes thinking patterns similarly to finetuning while avoiding the cost of parameter updates (Lin et al., 2024; Zhao et al., 2025; Yang et al., 2024), the question of *why* heavily engineered contexts often degrade model performance on complex real-world tasks—and how to systematically manage the resulting bottleneck—remains a critical challenge.

Concretely, aggressively engineered contexts suffer from a performance ceiling—recently termed *context rot* in engineering practice (Rajasekaran et al., 2025)—when applied to multi-faceted tasks requiring the integration of diverse information sources (He et al., 2024; Li et al., 2023b). This is distinct from simple long-context retrieval: while models can excel at needle-in-a-haystack style localization, integrating many densely interacting constraints still strains effective processing (Liu et al., 2024; Levy et al., 2024). Even when information fits easily within the physical limits of massive context windows, LLMs frequently suffer from attention diffusion, getting "lost in the middle" when forced to process dense, highly interactive context (Liu et al., 2024). The same bottleneck widens as agents increasingly rely on protocol-heavy tool usage and complex memory management (Packer et al., 2024). In such scenarios, when overwhelmed by extensive in-context information, LLM agents frequently exhibit degeneration of thought, hallucination of constraints, sycophantic agreement, or an inability to follow multiple interacting requirements (Liang et al., 2023; Huang et al., 2023; Kamoi et al., 2024; Lu et al., 2024; Li et al., 2025a). Despite increasing empirical studies on the limitations of long contexts and attention budgets (Xiao et al., 2023), the root causes remain underexplored. Concurrently, multi-agent LLM systems are increasingly popular for task distribution, yet many still coordinate through dense chat histories without explicit cognitive grounding, often exacerbating rather than mitigating context overload (Liu et al., 2023; Zhang et al., 2024c).

To address this gap, we turn to cognitive science for explanatory insight. Similar patterns of performance degradation under high informational demands have long been studied through the framework of Cognitive Load Theory (CLT) (Sweller, 2011; 2003). To better understand this phenomenon in LLMs, we first conduct a pilot study (Section 3). This study provides theoretical grounding by establishing the cognitive load framework for LLM performance limits and empirically examines this analogy through measurable proxies. Specifically, following CLT, we define an agent's **Working Memory (WM)** as its intrinsic, capacity-limited *attention budget*—the ability to simultaneously hold, filter, and process information active in its context window (Baddeley et al., 1986). Correspondingly, **Cognitive Load (CL)** is the demand that an engineered context places on an agent's WM, largely determined by the complexity and element interactivity of the prompt. When the CL imposed by a task exceeds the agent's WM capacity, a state of **cognitive overload** occurs. We conduct experiments measuring attention entropy and perplexity as proxies for context processing fluency and sparsity, demonstrating that LLMs exhibit patterns consistent with CLT predictions. Furthermore, recent evidence that LLMs exhibit bounded, human-like WM characteristics (Zhang et al., 2024b; Gong et al., 2024) supports reading the performance ceiling of heavy context engineering through a CLT lens: when the processing demands of in-context information approach the LLM's effective attention budget, performance degrades in ways *functionally analogous* to—not mechanistically identical to—the theoretical limits described by CLT. We treat CLT as a design lens yielding testable predictions about when multi-agent coordination helps (high intrinsic load) versus hurts (low intrinsic load), and evaluate both.

To translate these cognitive insights into actionable system design, we present *CoThinker*, a proof-of-concept multi-agent framework built to probe the utility of CLT as a design lens for managing context complexity. Instead of relying on heuristic coordination, *CoThinker* systematically operationalizes CLT principles through three key functions: (i) dynamic thinking style assignment that adapts to task demands, distributing the intrinsic cognitive load (task context complexity) across specialized agents, (ii) a transactive memory system that acts as *context compression* by maintaining shared knowledge, reducing the extraneous load of re-reading long raw histories, and (iii) a communication moderator that functions as *information routing*, limiting the influx of peer messages to protect each agent's attention budget while maintaining small-world network connectivity.

In sum, rather than merely proposing another multi-agent framework, this work provides a cognitive grounding for LLM coordination. Our key contributions are:

- We frame the failures of heavy context engineering through a CLT lens that maps human working memory constraints to LLM attention limits, supported by diagnostic empirical proxies (Section 3).
- We provide a principled, CLT-guided paradigm for multi-agent system design, instantiated through *CoThinker*, which operationalizes context compression, load distribution, and structured routing.
- We empirically evaluate CoThinker and find performance gains concentrated on reasoning-heavy, high-intrinsic-load tasks, with coordination overhead dominating on low-intrinsic-load instruction-following—consistent with the cognitive-load-profile prediction (Section 5.2).

## 2 Related Work

### 2.1 Multi-Agent LLM Collaboration

The rise of LLMs has spurred research into multi-agent systems (MAS), where LLMs collaborate to tackle complex problems beyond the scope of single agents (Guo et al., 2024; Wang et al., 2024a; Qian et al., 2025). Current approaches include multi-agent debates for idea exchange and critique (Liang et al., 2023; Lu et al., 2024; Wang et al., 2024b; Du et al., 2023), iterative reflection for self-correction (Shinn et al., 2023; Madaan et al., 2023; Yao et al., 2023), and functional specialization, where agents divide tasks in complex domains (Li et al., 2023a; Qian et al., 2023; Hong et al., 2023). Architecturally, research explores dynamic agent networks (Liu et al., 2023; Wu et al., 2023), mental set diversity (Liu et al., 2025b), and hierarchical coordination (Zhang et al., 2024a). However, these designs often rely on intuition or communication efficiency, with limited grounding in cognitive theories of processing constraints (Cemri et al., 2025). In fact, naive multi-agent communication often *increases* extraneous context load, as each agent is forced to append the raw, lengthy outputs of all its peers into its own context window. Our work, *CoThinker*, directly addresses this gap. By treating multi-agent coordination as a distributed context processing problem, *CoThinker* actively manages what information enters each agent's context window to prevent cognitive overload.

### 2.2 LLM for Human Simulation

The capacity of LLMs to exhibit human-like intelligence (Liu et al., 2025a) and emulate nuanced social behaviors (Zhou* et al., 2024) is foundational to their use as artificial agents. Research has demonstrated LLMs' ability to simulate human decision-making (Xie et al., 2024), generate believable individual and collective behaviors (Chuang et al., 2024a), and adopt distinct personas (Chuang et al., 2024b). Critically, these parallels extend to cognitive characteristics; recent studies suggest LLMs possess bounded working memory and exhibit failure modes under cognitive overload akin to humans (Zhang et al., 2024b; Gong et al., 2024). Furthermore, interactions between LLM agents can mirror social psychological phenomena (Zhang et al., 2024c; Guo et al., 2024). This confluence of human-like cognitive traits, including processing limitations and social capabilities, provides a strong rationale for applying Cognitive Load Theory to design more effective, load-aware collaborative LLM systems.

## 3 Cognitive Foundations for Enhanced LLM Performance

This section presents our pilot study, which establishes the theoretical foundation for our approach by linking human cognitive limitations to performance ceilings in LLMs. We introduce a cognitive load model based on working memory (WM) and attention budget analogies (Section 3.1), examine this analogy empirically (Section 3.2), and demonstrate how Cognitive Load Theory (CLT) can address individual limitations and guide LLM system design (Section 3.3).

### 3.1 A Cognitive Load Framework for LLM Performance Limits

As depicted in the first and second blocks of Fig. 1, we propose a model that draws parallels between human cognitive limitations and the context processing ceilings in LLMs.

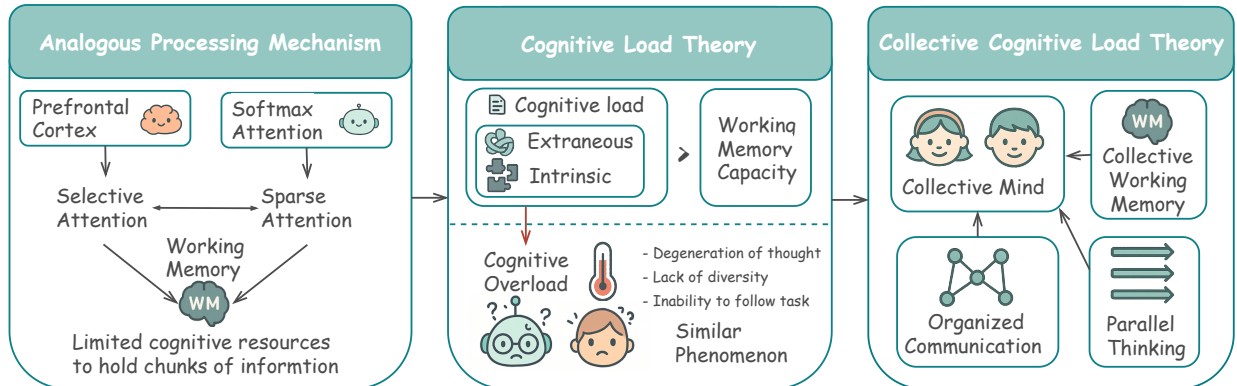

Figure 1: Cognitive Load framework: Using Cognitive Load Theory (CLT), we align human working memory constraints with LLM attention budgets to explain context rot in complex tasks and to guide distributed multi-agent methods to mitigate them.

**Analogous Processing Mechanisms.** Human cognition relies fundamentally on working memory, a capacity-limited cognitive system associated with the prefrontal cortex that employs selective attention to filter and prioritize information during complex cognitive tasks (Baddeley et al., 1986; Cowan, 2010; Miller, 1956). LLMs exhibit structural parallels we treat as a design analogy rather than a mechanistic claim about transformer internals: their softmax attention mechanisms perform selective focus on input data (Vaswani et al., 2017), with attention heads specializing in distinct processing patterns (Voita et al., 2019). Crucially, while modern LLMs possess massive *physical* context windows, their *active processing capacity*—their attention budget—remains strictly bounded. Recent studies provide direct evidence that LLMs possess human-like WM characteristics, exhibiting clear limitations on concurrent information processing with performance degrading predictably as cognitive demands increase (Zhang et al., 2024b; Gong et al., 2024). More discussion in Appendix B.1.1, B.1.2.

**Cognitive Load Theory.** Building upon these working memory analogies, we apply CLT (Sweller et al., 1998; Sweller, 2011) to interpret LLM performance patterns under heavy context engineering. CLT distinguishes Cognitive Load (CL) between *intrinsic load* (determined by task complexity and element interactivity) and *extraneous load* (arising from how instructions and information are presented). When the combined load exceeds working memory capacity, *cognitive overload* ensues. LLM agents demonstrate analogous performance degradation when tasked with complex problems via dense context engineering: tasks requiring extensive multi-step reasoning, exhaustive tool constraints, or long multi-agent dialogue histories can lead to degeneration of thought, lack of diversity, or inability to follow multiple requirements (Liang et al., 2023; Huang et al., 2023; Kamoi et al., 2024; Lu et al., 2024). We contend that such performance ceilings represent cognitive overload, where the total context processing demands surpass the LLM's effective attention budget. Examples in Appendix B.1.4.

## 3.2 Understanding Cognitive Load and Working Memory in LLMs

We empirically examine the analogy of CL and WM in LLMs. We probe into measurable proxies for cognitive load effects by definition and examine key CLT predictions regarding task and instruction complexity effects. By definition, WM handles information processing, and cognitive load represents the *attention* required to handle information within WM, which determines the *easiness* of task completion. We identified two proxies corresponding to these key characteristics:

**Attention Entropy** measures the diversity of the model's attention distribution. Higher entropy indicates more distributed attention across input tokens, suggesting the model must consider multiple aspects of the engineered context simultaneously, corresponding to a saturated attention budget and higher cognitive load (Zhang et al., 2025). **Perplexity** measures the model's certainty of solutions, serving as a proxy for the *fluency* of context processing. Both proxies are diagnostic rather than runtime signals—perplexity requires

ground-truth answers and attention entropy is model-internal—and we use them as supporting evidence for the analogy rather than as operational controls; identifying inference-time cognitive-load metrics that work without ground truth remains an open direction (App. B.1.1). For the *Task Complexity Effect* experiment, we construct Q&A pairs from AMPS (Hendrycks et al., 2021) with 4 difficulty levels (simple to complex arithmetics), controlling input length for fair comparison. For the *Instruction Complexity Effect* experiment, we select Q&A pairs from FLASK (Ye et al., 2023) with varying instruction complexity, measuring perplexity on answers for both hard and easy tasks. See Appendix B.2 for details of the experimental setup, results, and discussion of these proxies.

Table 1: Pilot study results: (Left) Attention entropy increases with task complexity, indicating higher cognitive load. (Right) Perplexity patterns are aligned with CLT predictions: instructions help reduce cognitive load for hard tasks but show no benefit for easy tasks.

**Experiment 1: Task Complexity Effect.**

| Task | Attention Entropy |
|---|---|
| Level 1 | 4.44 |
| Level 2 | 4.80 |
| Level 3 | 5.04 |

**Experiment 2: Instruction Complexity Effect.**

| Instruction | PPL (Hard) | PPL (Easy) |
|---|---|---|
| Level 1 | 120.50 | 3.37 |
| Level 2 | 88.97 | 3.42 |
| Level 3 | 85.35 | 3.45 |

The results are consistent with our CLT-LLM analogy. Attention entropy increases with task complexity (Table 1, left), which is consistent with harder tasks requiring the model to process more information chunks simultaneously and saturating its attention budget. For perplexity (Table 1, right), hard tasks show decreasing perplexity with instruction complexity, which suggests well-engineered instructions help the model focus and reduce cognitive load. Easy tasks show increasing perplexity, indicating overly complex instructions provide no benefit and may introduce extraneous load—a pattern matching CLT's redundancy effect where additional context impairs performance when task demands are already within capacity. We treat these proxy patterns as design-time evidence for the analogy rather than proof of a CLT mechanism inside the model.

### 3.3 Collective Intelligence Principles for Cognitive Load Management

With this proxy evidence in hand, we can now leverage CLT principles to address context rot and cognitive overload in LLMs. As depicted in the final block of Fig. 1, when humans encounter tasks that exceed individual WM capacity, we employ two strategies: external tools or collective intelligence. For complex tasks where external tools are insufficient, humans naturally form collaborative cognitive systems that exceed individual capabilities, leading to the emergence of a *collective mind* (Woolley et al., 2010; Malone et al., 2010). CLT provides principled guidance for managing cognitive load within such collective systems, particularly addressing how the introduction of new agents or coordination mechanisms can introduce extraneous load that must be carefully balanced. Appendix B.1.3 provides discussion from cognitive science.

This collective intelligence effectively manages cognitive load through three core mechanisms guided by CLT principles: (i) **Division of Cognitive Labor** through parallel thinking, allowing individuals to focus on specialized aspects of problems, thereby reducing intrinsic load per individual (Dunbar, 2003); (ii) **Collective Working Memory**, often through Transactive Memory Systems where knowledge and responsibilities are distributed. This acts as a natural form of context compression, enabling individuals to rely on each other for information sharing while managing the extraneous load of coordination (Wegner, 1987; Kirschner et al., 2018); and (iii) **Structured Communication** that efficiently integrates diverse insights through organized information flow, preventing each individual's attention budget from being overwhelmed by coordination overhead (Hutchins, 1995). Since LLMs face analogous attention limitations and our pilot study indicates measurable context load effects, systematically operationalizing these CLT principles into a multi-agent architecture is a promising route to bypass the single-agent context bottleneck.

## 4    CoThinker: Instantiating CLT in Multi-Agent Design

*CoThinker* is not merely a conversational framework, but a structural instantiation of the CLT principles and collective intelligence mechanisms outlined in Section 3. Simply aggregating outputs from LLM agents often proves insufficient for complex tasks, as naive collaboration introduces massive *transactional costs*—forcing each agent to append raw, lengthy peer outputs into its own context window. As CLT predicts, this extraneous load quickly leads to cognitive overload, negating the benefits of parallel computing (Kirschner et al., 2009; 2018; Cemri et al., 2025). *CoThinker* translates cognitive insights into a practical architecture to explicitly distribute and compress context load rather than scaling collaboration by brute-force context expansion.

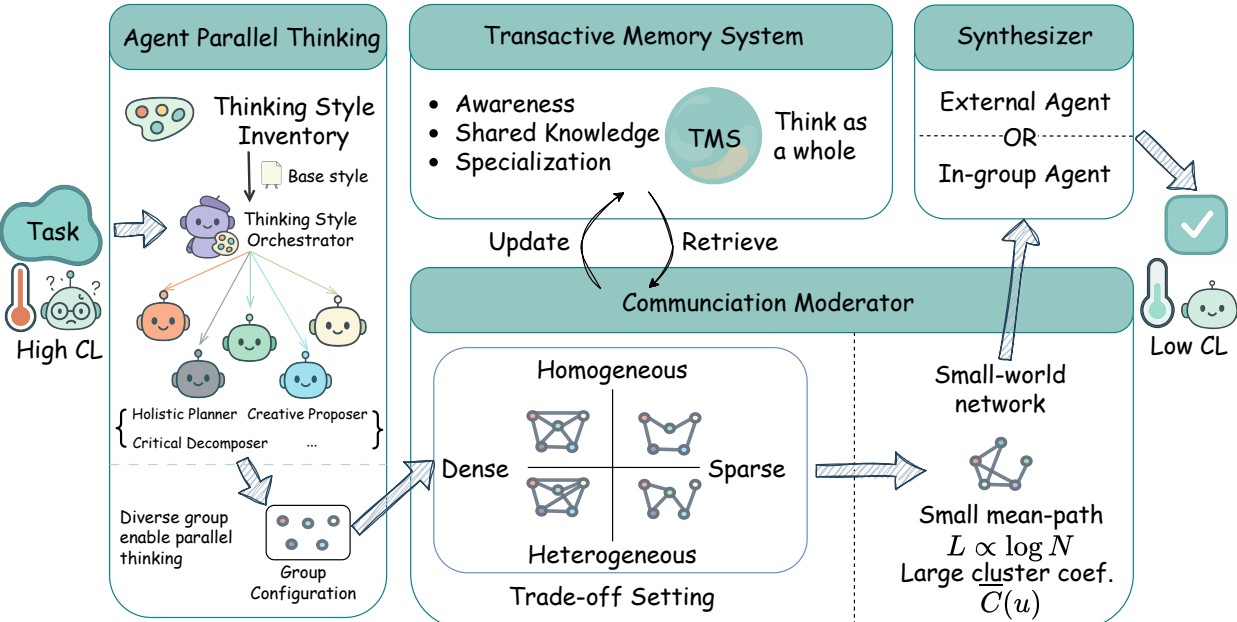

Figure 2: The *CoThinker* Architecture. A high CL task is initially processed by diverse agents via Agent Parallel Thinking. Transactive Memory System facilitates shared understanding by updating and retrieving collective knowledge. Communication Moderator manages inter-agent information flow, leveraging a trade-off to form a cognitive small-world network, which then feeds into the Synthesizer for final solution, resulting in a lower effective CL for the overall system.

To operationalize these insights, the *CoThinker* architecture (Figure 2) comprises four main modules: Agent Parallel Thinking (Section 4.1), Transactive Memory System (Section 4.2), Communication Moderator (Section 4.3), and Synthesizer (Section 4.4). Each module is directly guided by CLT principles to emulate aspects of the human collective mind. Agent Parallel Thinking addresses intrinsic load through division of cognitive labor. The Transactive Memory System provides context compression, reducing extraneous load from redundant raw history. The Communication Moderator enforces information routing and caps peer influx into each agent's context. Finally, the Synthesizer integrates refined collective insights. Let $\mathcal{A} = \{A_1, \ldots, A_M\}$ be the set of $M$ agents. Let $T_{\max}$ be the total number of generation rounds. Agent $A_i$'s output at the end of round $t$ is denoted $x_i^{(t)}$.

### 4.1    Agent Parallel Thinking

This module addresses **intrinsic cognitive load** by promoting a *division of cognitive labor*: each agent focuses its limited attention budget on a task-specific thinking dimension rather than juggling all aspects of the problem in one monolithic prompt. Unlike assigning pre-defined roles, which require domain-specific foresight and impose extraneous CL from role adherence, *CoThinker* uses an adaptive approach. A Thinking Style Orchestrator generates a task-specific style $\phi_i$ for each agent $A_i$ based on a general base thinking style

inventory $\psi$ (Sternberg, 1997) and the task $D$:

$$\{\phi_i\}_{i=1}^M = \text{Orch}(D, \psi) \tag{1}$$

This yields diverse thinking styles $\{\phi_i\}_{i=1}^M$, employed in subsequent stages. Unlike pre-defined roles requiring complex persona maintenance (extraneous load), thinking styles (Sternberg, 1997) represent preferred ways of applying capabilities, enabling division of labor with minimal overhead. Further details on the prompting strategy for style generation and thinking style inventory are in Appendix D.3.

### 4.2 Transactive Memory System (TMS)

This module addresses **extraneous cognitive load** through **context compression**. Human groups effectively manage complex information by developing Transactive Memory Systems (TMS), which involve a shared understanding of who knows what, how to access information held by others, and a collective agreement on the information itself (Wegner, 1987; Hollingshead, 2001). This distributed cognitive system allows individuals to specialize and rely on others, reducing individual CL and enhancing group problem-solving (Lewis, 2003). In *CoThinker*, TMS replaces the need for every agent to re-read full raw chat transcripts each round: an evolving structured summary $\mu^{(t)}$ distills consensus, expertise cues, and open issues into a compact state, acting as an information bottleneck that limits token growth as rounds progress. Implementation details in Appendix D.4. At each round $t$, the group's collective knowledge representation $\mu^{(t)}$ is updated based on contributions from all agents:

$$\mu^{(t+1)} = \text{UpdateMem}(\mu^{(t)}, \{x_j^{(t)}\}_{j=1}^M) \tag{2}$$

### 4.3 Communication Moderator

This module further manages extraneous load via **information routing**. Effective inter-agent communication is crucial, yet it incurs transactional costs—the cognitive effort for message processing and integration—imposing extra extraneous CL. The moderator selects only $N < M$ reference messages for each agent $A_i$, hard-capping how much peer content enters that agent's context and protecting its attention budget from unbounded broadcast chatter. This process navigates the critical trade-offs between **Network Density vs. Sparsity** (high exposure vs. information loss) and **Information Homogeneity vs. Heterogeneity**. The latter involves balancing the ease of integrating cognitively similar inputs (low extraneous load but risk of echo chambers (Runkel, 1956)) against the benefits of diverse perspectives for distributing intrinsic load (Aral & Van Alstyne, 2011).

**Communication Topology and Algorithm:** The selection of references defines a directed communication graph $G^{(t-1)} = (\mathcal{A}, E^{(t-1)})$ for each round, where an edge $(A_u, A_v) \in E^{(t-1)}$ exists if agent $A_v$ receives a message from agent $A_u$ generated in round $t-1$. Motivated by how small-world networks efficiently balance local clustering with global connectivity (Watts & Strogatz, 1998), our moderator employs the following algorithm to construct this graph:

a. **Fixed In-Degree ($N$):** Each agent $A_i$ (node $A_v$) has an in-degree of $N$, capping its processing load and respecting LLM WM (Zhang et al., 2024b; Gong et al., 2024).
b. **Define Cognitive Distance between Agent Outputs:** The cognitive distance $d(x_u^{(t-1)}, x_v^{(t-1)}) = 1 - \text{sim}(x_u^{(t-1)}, x_v^{(t-1)})$ is based on the semantic similarity of previous outputs.
c. **Re-connection via Probabilistic Rewiring ($\beta$):** For each agent $A_i$, its $N$ incoming edges (references $\mathcal{P}_i^{(t-1)}$) are chosen from cognitively similar peers (low distance), but with a probability $\beta$, "rewiring" some connections to randomly chosen, diverse peers.

**Resulting Network Properties and Cognitive Balance:** It fosters dynamic communication networks with small-world properties, with high local clustering (facilitating efficient refinement of similar ideas, reducing extraneous load locally) and short average path lengths (enabling rapid global propagation of diverse insights, aiding intrinsic load distribution). This structure offers a balance between focused collaboration and broad information access, managing CL more effectively than random or regular lattice networks. Further details are in Appendix D.5.

### 4.4 Synthesizer

The Synthesizer consolidates all agents' answers and TMS into a final answer (details in Appendix D.6). ***CoThinker* Process Flow.** The process for task $D$ with $M$ agents over $T$ rounds:

*Initialization:*

$$\{\phi_i\}_{i=1}^M = \text{Orch}(D, \psi_i), \quad x_i^{(0)} = \text{Agent}(D, \phi_i), \quad \mu^{(0)} = \text{UpdateMem}(\{x_i^{(0)}\}_{i=1}^M) \tag{3}$$

*Iterative Refinement for agent $A_i$ and round $t$:*

$$\mathcal{P}_i^{(t)} = \text{SelectRefs}(\{x_k^{(t)}\}, N, \beta) \tag{4}$$

$$x_i^{(t+1)} = \text{Agent}(D, \phi_i, \mu^{(t)}, x_i^{(t)}, \mathcal{P}_i^{(t)}), \quad \mu^{(t+1)} = \text{UpdateMem}(\mu^{(t)}, \{x_k^{(t+1)}\}) \tag{5}$$

*Final Synthesis:*

$$y_{\text{final}} = \text{Synth}(\{x_i^{(T-1)}\}_{i=1}^M, \mu^{(T-1)}, D) \tag{6}$$

Compared to existing multi-agent debate frameworks that scale by adding agents and broadcasting peer outputs into every agent's context (Du et al., 2023; Liang et al., 2023; Liu et al., 2025b), three choices in the design above follow directly from the CLT lens: thinking styles are generated adaptively for each task rather than fixed as personas (so role adherence consumes less of the attention budget), peer input is routed through a moderator that caps how much each agent must integrate per round, and the resulting system is expected to help only where intrinsic load is high enough to justify the coordination cost—a task-dependent boundary the "more agents are better" view does not predict and which our experiments are consistent with in Section 5.2.

## 5 Experiments and Results

This section details our experimental methodology and empirically evaluates the CLT-driven design instantiated in *CoThinker*. We outline the experimental setup, present main results on LiveBench (White et al., 2025) and CommonGen-Hard (Madaan et al., 2023), followed by ablations and discussion through the lens of Cognitive Load Theory (CLT).

### 5.1 Experimental Setup

**Models and Configuration.** For main experiments, we use three Gemini models (Team et al., 2024) with varying capacities: Gemini-1.5-Flash-8B (lightweight), Gemini-1.5-Flash (mid-tier), and Gemini-1.5-Pro (high-capacity). **Evaluation Benchmarks.** We evaluate on two challenging benchmarks: (1) *LiveBench* (White et al., 2025), a broad-coverage benchmark of *real-world tasks* with periodic task updates, providing evaluation across math, reasoning, data analysis, and related domains; and (2) *CommonGen-Hard* (Madaan et al., 2023), a controlled *experimental challenge* designed to test information interactivity, by forcing models to integrate target concepts from large pools of distractors. **Baselines.** We compare *CoThinker* with both single-agent and multi-agent approaches: Single Agent (IO), Single Agent (CoT) (Wei et al., 2022), Single Agent (Self-Refine) (Madaan et al., 2023), Multi-Agent Debate (MAD) (Du et al., 2023; Liang et al., 2023), and Diverse MAD (DMAD) (Liu et al., 2025b). Complete details are in Appendices E.2, and E.1.

### 5.2 Main Results on LiveBench

Table 2 reports *CoThinker* accuracy on LiveBench across three Gemini models alongside five baselines. Since we use greedy decoding (temperature 0), so sampling variance will be near zero; we therefore report a bootstrap SE per cell ($B$=2,000, within-subtask resampling) capturing question-sampling variance, rather than a sampling-CI. The bootstrap SE and the paired Holm tests it uses are stricter than naive sampling-based intervals. To assess significance we run paired Holm–Bonferroni tests on per-question scores under two families (vs. single-agent: 15 tests; vs. multi-agent: 10 tests; App. C.2, Table 13). Sixteen of 25 comparisons

Table 2: LiveBench results per (model, category, method). Each cell shows three lines: aggregated task score, per-subtask ratio normalized to Flash-8B IO, and bootstrap standard error. **Bold** marks the highest score per (model, category) block.

| | Gemini-1.5-Flash-8B | | | | | | Gemini-1.5-Flash | | | | | | Gemini-1.5-Pro | | | | | |
|---|---|---|---|---|---|---|---|---|---|---|---|---|---|---|---|---|---|---|
| **Task** | IO | CoT | SR | MAD | DMAD | Ours | IO | CoT | SR | MAD | DMAD | Ours | IO | CoT | SR | MAD | DMAD | Ours |
| Math | 33.8 [1.00] (±2.5) | 34.1 [1.04] (±2.6) | 31.7 [0.92] (±2.5) | **38.6** **[1.13]** (±**2.6**) | 35.2 [1.13] (±2.8) | 37.7 [1.11] (±2.7) | 48.6 [1.47] (±2.8) | 48.6 [1.47] (±2.8) | 47.6 [1.45] (±2.8) | 49.5 [1.51] (±2.8) | 49.2 [1.49] (±2.7) | **51.2** **[1.57]** (±**2.9**) | 60.9 [2.00] (±2.6) | 62.4 [1.86] (±2.7) | 61.4 [1.93] (±2.6) | 69.4 [2.29] (±2.3) | 70.0 [2.31] (±2.4) | **75.3** **[2.40]** (±**3.7**) |
| Data | 14.0 [1.00] (±1.9) | 12.6 [0.90] (±2.0) | 4.9 [0.34] (±0.7) | 8.1 [0.58] (±1.8) | 8.9 [0.64] (±1.4) | **18.4** **[1.32]** (±**3.1**) | 25.5 [2.03] (±3.0) | 26.4 [2.07] (±3.3) | 18.0 [0.90] (±3.3) | 21.9 [1.46] (±2.9) | **27.6** **[2.51]** (±**3.5**) | 27.5 [2.44] (±3.4) | 40.7 [2.92] (±4.7) | 38.0 [2.72] (±4.3) | 18.6 [1.33] (±3.5) | 43.9 [3.15] (±4.5) | 46.4 [3.32] (±4.4) | **47.3** **[3.39]** (±**4.8**) |
| Reas. | 23.5 [1.00] (±3.1) | 20.9 [1.11] (±3.0) | 15.8 [0.80] (±2.7) | 24.8 [1.21] (±3.1) | 17.1 [0.85] (±2.9) | **28.5** **[1.22]** (±**3.4**) | 38.3 [1.63] (±3.4) | 41.0 [1.74] (±3.4) | 36.5 [1.55] (±3.3) | 45.0 [1.92] (±3.5) | 44.9 [1.94] (±3.5) | **45.3** **[1.97]** (±**3.5**) | 43.0 [1.87] (±3.8) | 42.0 [1.82] (±3.8) | 36.9 [1.80] (±3.6) | 40.9 [1.78] (±3.7) | 42.7 [1.88] (±3.8) | **44.7** **[1.95]** (±**3.7**) |
| Lang. | 16.1 [1.00] (±2.8) | 19.4 [1.09] (±3.1) | 18.0 [0.89] (±3.2) | 20.1 [1.03] (±3.3) | 26.5 [1.02] (±3.7) | **29.4** **[0.98]** (±**3.9**) | 32.2 [1.41] (±3.9) | 29.1 [1.30] (±3.8) | 28.7 [1.06] (±3.7) | 33.0 [1.46] (±4.0) | 36.2 [1.44] (±3.8) | **40.2** **[1.52]** (±**4.1**) | 37.4 [1.43] (±3.8) | 34.9 [1.54] (±3.8) | 42.8 [1.22] (±3.9) | 46.1 [1.58] (±3.4) | 52.1 [1.74] (±4.1) | **52.6** **[1.76]** (±**3.9**) |
| Instr. | **84.3** **[1.00]** (±**1.5**) | 83.5 [1.02] (±1.6) | 69.9 [0.81] (±2.0) | 78.1 [0.87] (±1.9) | 78.6 [0.89] (±1.9) | 68.1 [0.80] (±3.4) | 87.1 [1.10] (±1.5) | 87.1 [1.10] (±1.6) | 73.4 [0.87] (±2.0) | 81.4 [1.01] (±1.9) | **87.3** **[1.06]** (±**1.6**) | 80.2 [0.99] (±2.9) | **84.4** **[1.03]** (±**1.2**) | 83.6 [1.02] (±1.3) | 61.5 [0.72] (±1.6) | 67.6 [0.77] (±1.6) | 83.1 [1.02] (±1.3) | 81.8 [0.95] (±1.3) |

clear Holm at $\alpha$=0.05: *CoThinker* beats every single-agent baseline and MAD on most tasks. Several near-significant comparisons (e.g., vs. DMAD on Reasoning and Data Analysis) miss Holm at $\alpha$=0.05 even though their point estimates are directionally favorable and reproducible under our near greedy setting; this is a consequence of the stricter test rather than score instability. The three Holm-significant losses all fall on instruction-following, the CLT-predicted low-intrinsic-load boundary.

Concretely, the *cognitive-load profile* sorts a task by how much working-memory pressure it imposes before any coordination overhead: **high-intrinsic-load** tasks are reasoning-heavy and decomposable into distinct sub-problems (Math, Data Analysis, Reasoning, Language), while **low-intrinsic-load** tasks are execution-focused and rubric-following (Instruction-Following), so a single agent already handles them well. **High intrinsic CL tasks** scale with model capability and benefit from distributing load across agents through moderated routing and shared memory, while **low intrinsic CL tasks** gain little from stronger models and *CoThinker*'s coordination overhead introduces extraneous CL that outweighs the collaboration benefit. This split is the boundary the CLT lens predicts: coordination helps where intrinsic load is high and hurts where extraneous overhead dominates. Details appear in Appendix E.

### 5.3 Main Results on CommonGen-Hard

*CoThinker* shows measurable improvements on CommonGen-Hard, a task designed to probe high-CL management. Figure 3 shows performance across evaluation dimensions through (a) a radar plot with normalized by dividing the min scores and (b) an interaction rounds plot tracking performance evolution. Relative to heuristic multi-agent debate-style protocols, the CLT instantiation in *CoThinker*—partitioning intrinsic load, compressing shared state, and routing peer input—better preserves reasoning quality as rounds accumulate. The radar plot (Figure 3a) reveals strengths in coherence and concept integration, with minor trade-offs in conciseness. The rounds plot (Figure 3b) shows sustained improvement across multiple interaction rounds: baseline methods degrade as unstructured coordination overhead (extraneous CL) accumulates.

### 5.4 Cross-Model Generalization

To check that the observed gains are not Gemini-specific, we evaluated across multiple LLM families: GPT-5-Nano, Qwen3-30B-A3B, GPT-OSS-20B, Gemini-2.5-Flash, GPT-4.1-Mini, Qwen3-32B, and DeepSeek-R1-8B (partial results in Table 3). We examine two scenarios: (1) standard setting with IO baselines using

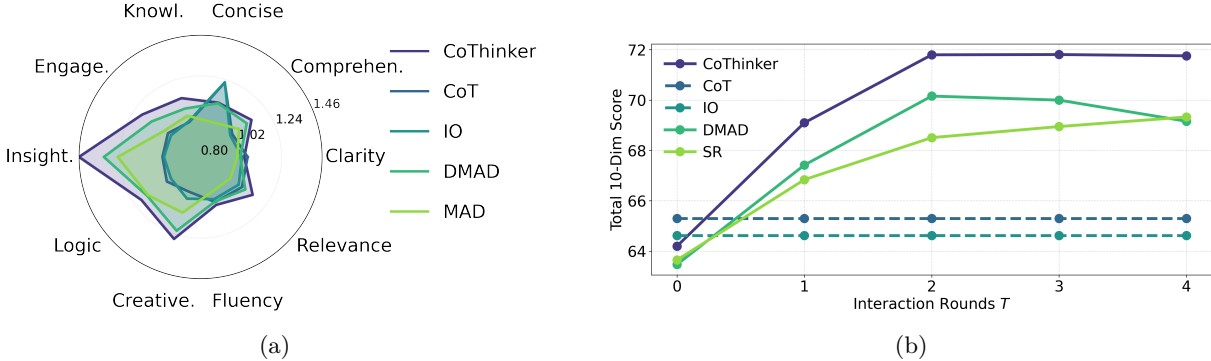

(a)                                                                 (b)

Figure 3: *CoThinker* performance on CommonGen-Hard (Madaan et al., 2023) using Gemini-1.5-Flash. (a) The radar plot illustrates a multi-dimensional performance, showing well-rounded improvement. (b) The rounds plot depicts the total score across rounds, showing stable improvement.

maximum reasoning steps with temperature 0.25 and (2) constrained setting with token budget of 8192 with greedy decoding (temperature = 0). Complete results are in Appendix E.3.

Table 3: Cross-model evaluation on LiveBench (White et al., 2025) Math and Reasoning (Reason.) subsets in standard setting (left) and constrained setting (right).

| | Standard Setting | | |
|---|---|---|---|
| Model | Method | Math | Reason. |
| GPT-5 | CoThinker | **88.57** | **81.88** |
| | IO | 82.63 | 68.38 |
| Qwen3 | CoThinker | **80.62** | **89.50** |
| | IO | 77.50 | 76.00 |

| | Constrained Setting | | |
|---|---|---|---|
| Model | Method | Math | Reason. |
| Gemini-2.5 | CoThinker | **76.3** | **69.2** |
| | IO | 59.3 | 31.0 |
| GPT-4.1 | CoThinker | **40.0** | **70.8** |
| | IO | 34.0 | 40.8 |

## 5.5 CoThinker Ablation of Communication Moderator

We ablate the Communication Moderator's key parameters—reference set size ($N$), exploration rate ($\beta$), and agent count ($M$)—on Gemini-1.5-Flash-8B across four LiveBench categories: Math, Reasoning, Data Analysis, and Instruction (See Figure 4). Default parameters: $T = 3$, with controlled variations for each parameter. Scores are normalized by IO baseline. **Analysis.** The Communication Moderator's parameters directly control CL balance: *Reference set size (N)* manages extraneous load—optimal $N = 2-3$ balances peer input diversity against overload, respecting LLM working memory limits. *Exploration rate ($\beta$)* governs similarity-diversity trade-offs: low $\beta$ (exploiting similar ideas) reduces integration load but risks echo chambers; high $\beta$ (exploring diverse perspectives) aids intrinsic load distribution but increases extraneous load. Task-dependent optima (e.g., higher $\beta$ for Reasoning) reflect this balance through small-world network properties. *Agent count (M)* shows non-monotonic performance—more agents distribute intrinsic load but elevate coordination costs, a pattern consistent with CLT predictions for group overload. These findings support the Communication Moderator's role in managing CL for effective collective intelligence. See Appendix E.5 for details.

## 5.6 CoThinker Ablation on Other Components

We conducted component ablation on Transactive Memory System (TMS) and Thinking Style Orchestrator (Style) a subset of LiveBench Math tasks. We also examine the proxy, perplexity (PPL), to reflect the CL management effects of our components.

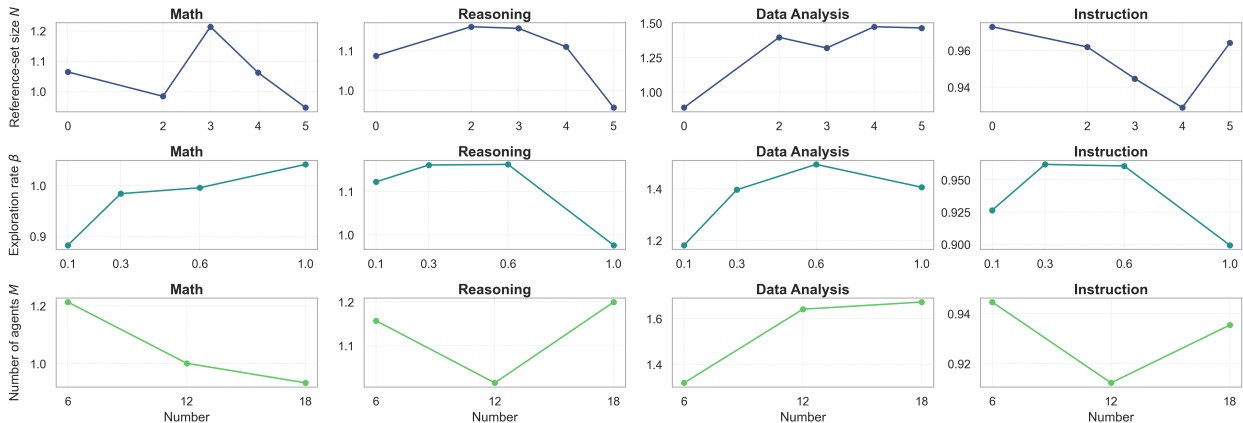

Figure 4: Communication Moderator ablation on *CoThinker* using Gemini-1.5-Flash-8B. **Top**: Reference Set Size ($N \in \{0, 2, 3, 4, 5\}$, $M = 6$, $\beta = 0.3$); **Middle**: Exploration Rate ($\beta \in \{0.1, 0.3, 0.6, 1.0\}$, $N = 2$, $M = 6$); **Bottom**: Agent Count ($M \in \{6, 12, 18\}$, $N = 3$, $\beta = 0.3$).

**Component Ablation.** Communication Moderator is fixed ON for all runs. We test configurations: *TMS* $\in \{On, Off\}$; *Style* $\in \{On, Off\}$. **Analysis.** Our components benefit most models (details in Appendix E.4). From Table 4, we find Thinking Style Orchestrator provides consistent improvements. However, TMS is less effective for new GPT models. Investigating their output, we find that they often refuse to give intermediate results, responding with "I can't provide step-by-step reasoning." This is counterproductive, as it discourages the detailed reasoning used to build TMS. This is a practical deployment consideration: backbones that suppress intermediate reasoning traces (common in newer safety-tuned models) degrade TMS effectiveness, and the right fix is to design TMS prompts that synthesise from the agents' final-answer content rather than asking the backbone to expose its reasoning, since the latter triggers the suppression behaviour.

Table 4: Component ablation on subset of Math dataset, with effect of TMS and Thinking Style Orchestrator, indicating the CL management benefits of each component.

| Configuration | Qwen3-30B-A3B | GPT5-Nano | GPT-OSS-20B |
|---|---|---|---|
| TMS: ON, Styles: ON | 81.87 | 55.97 | 57.15 |
| TMS: ON, Styles: OFF | 69.79 | 49.02 | 48.21 |
| TMS: OFF, Styles: ON | 76.41 | 62.36 | 58.37 |

**PPL proxy as Evidence.**. We conducted perplexity (PPL) studies on weaker models to demonstrate how our components helps weaker models reduce CL in understanding stronger models' outputs. We choose Math and Reasoning tasks for our analysis. Lower PPL indicates higher easiness and effective CL reduction (See Table 5). For more interesting ablation with PPL proxy, see Appendix E.4.

Table 5: PPL ablation showing CL reduction effects: how components help weaker models process information from better models' answers, reducing CL (lower PPL).

| Model | Baseline | Styles | TMS | References (N=3) |
|---|---|---|---|---|
| Qwen3-8B | 6.56 | 3.58 | 1.69 | 3.10 |
| Mistral-7B | 6.63 | 5.04 | 1.58 | 2.86 |

### 5.7 Cost and Latency

Output tokens per question are reported in Table 6. *CoThinker* uses output tokens on the same order as MAD/DMAD on every backbone (within 0.89–1.11× of MAD). The multi-agent overhead is worth paying on reasoning-heavy queries, where the accuracy gain is largest, but not on low-intrinsic-load tasks, so the cognitive-load profile (Section 5.2) doubles as a deployment-time routing signal.

Table 6: Inference cost per question on the high-intrinsic-load LiveBench subset (estimation protocol in App. E.9).

| Metric | Model | IO | CoT | SR | MAD | DMAD | Ours |
|---|---|---|---|---|---|---|---|
| Tokens/q | Flash-8B | 706 | 619 | 2,086 | 12,389 | 10,482 | **13,688** |
| | Flash | 638 | 643 | 2,566 | 16,104 | 13,688 | **14,323** |
| | Pro | 660 | 666 | 2,521 | 14,116 | 12,513 | **13,558** |
| API calls/q | *all* | 1 | 1 | 3 | 18 | 18 | **~25** |

## 6 Conclusion

This work interprets the performance ceilings of heavy context engineering through a Cognitive Load Theory (CLT) lens, treating the CLT-LLM correspondence as a functional analogy rather than a mechanistic claim. We draw parallels between human working memory constraints and LLM attention budgets, with diagnostic load proxies for support. More importantly, we describe how these theoretical insights can systematically guide system design. By instantiating *CoThinker*—a framework that utilizes agent specialization, context compression (TMS), and information routing—we observe that mitigating intrinsic and extraneous load improves collective problem-solving over heuristic multi-agent baselines on reasoning-heavy tasks, while coordination overhead dominates on low-intrinsic-load instruction-following tasks—a boundary the cognitive-load-profile view predicts (Section 5.2). Our findings suggest that as LLM agent systems scale in complexity, relying on heuristic prompt engineering and unstructured chat protocols is insufficient. Future multi-agent architectures should embrace cognitively grounded, load-managed designs to fully unlock the collaborative potential of large language models. **When to reach for CoThinker.** A rough practitioner heuristic: (i) single-agent CoT suffices when intrinsic load is low (e.g., instruction-following with a clear rubric); (ii) multi-agent coordination pays off when the task decomposes into distinct reasoning styles or sub-problems; (iii) the moderator matters most when raw peer output would otherwise dominate each agent's context window.

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

# Appendix

## A  Catalog of supplementary material

This supplement holds material referenced from the main text: extended theory, statistics, implementation detail, auxiliary results, and extended discussion (limitations, ethics, reproducibility, and future work).

**Appendix B is the central appendix.** It does three jobs in one place. First, it expands the cognitive foundations and the CLT-to-LLM analogy from Section 3.1: human working memory and attention, parametric knowledge vs. in-context processing, mechanisms of collective intelligence, and short CLT-based readings of representative LLM failure modes from the literature. Second, it documents the *pilot study* that motivates attention entropy and perplexity as load proxies—formal definitions, theoretical motivation, Mistral-7B protocol (including prefilled-answer perplexity), two controlled experiments (task complexity via AMPS-style difficulty; instruction complexity via FLASK), tables for both proxies, and an explicit statement that these measures are diagnostic rather than operational at test time. Third, it gives post-hoc perplexity analyses on the main CoThinker experiments (component ablations, reference count $k$, and peer-selection strategy), with the full PPL tables that Section 5.6 summarizes.

Appendix C gives small-world statistics on moderated communication graphs and bootstrap standard-error reporting for network- and score-level claims.

Appendix D compresses CoThinker's message flow, comparison to other multi-agent stacks, and the actual prompt shells for parallel thinking, TMS, the moderator's network rationale, and the external synthesizer.

Appendix E records API defaults, baseline definitions, LiveBench and CommonGen-Hard scope, and the spillover tables and figures behind the main experiments (Section 5.1 and results).

Appendix F gives extended discussion in two parts—ethics and reproducibility, then limitations and future directions—so the main conclusion stays short.

## B  Theoretical Foundations and Validation

### B.1  Cognitive Foundations and the CLT-LLM Analogy

Here we elaborate on the cognitive science foundations underpinning our framework and the analogy to LLMs from Section 3.1.

#### B.1.1  Human Working Memory and Attentional Control

Human working memory (WM) is a core cognitive faculty for actively holding and manipulating a limited amount of information relevant to ongoing tasks, operating through attentional mechanisms that select and maintain internal representations, often associated with sustained neural activity in regions like the prefrontal cortex (Baddeley et al., 1986; Cowan, 2010; Postle, 2006). Given that Large Language Models exhibit emergent sparse attention—where specific attention heads specialize in processing distinct patterns rather than diffusely attending to all input tokens (Vaswani et al., 2017; Voita et al., 2019)—it prompts an intriguing question: does this selective information processing within a finite context window imply the existence of a functional analogue to human WM in LLMs? This emergent selectivity, where not all information in the context is equally weighted or actively processed at any given step, forms a crucial part of the analogy we draw to understand potential capacity limitations and cognitive load phenomena in these models, particularly when handling tasks with high element interactivity under dense engineered context.

#### B.1.2  Working Memory and Parametric Knowledge in LLMs

**Misunderstanding 1: Working Memory as Context Window Limits** A prevalent misunderstanding in current LLM research equates Working Memory (WM) with physical limitations like context window size and assumes that models only fail when they exceed their maximum token limits—that WM is simply the amount of in-context information that can fit within the context window. However, our framework

distinguishes WM as the mechanism of selective attention: a limited cognitive capacity to actively hold and manipulate information simultaneously, not merely passive storage of tokens. This selective attention mechanism can become overwhelmed and lead to performance degradation well before reaching physical context limits.

**Misunderstanding 2: High Cognitive Load as Long Input/Output Tasks** Another common misconception is that high cognitive load (CL) tasks are simply those with long inputs and long outputs. This conflates task length with cognitive complexity. In our framework, a high CL task is defined by the working memory demands it places on simultaneous information processing—specifically, the amount of information that must be actively held and manipulated *simultaneously* in WM. For example, a sequential fact elicitation task (e.g., "List 100 countries and their capitals") may have long input and output but requires minimal WM since each fact can be retrieved independently without integrating interdependent information. Conversely, a complex reasoning task requiring simultaneous consideration of multiple interacting constraints, relationships, or variables creates high CL regardless of input/output length, as it demands that the model maintain and manipulate multiple information pieces concurrently in its working memory.

**Our Framework: Working Memory and Parametric Knowledge** Our framework clarifies that an LLM's parameters can be understood as 1)*Parametric Knowledge* (analogous to human long-term memory, mainly from FFN layer Geva et al. (2021)), and 2) innate generation capabilities that determine how effectively the model can select and process information for its WM (mainly attention layer Tighidet et al. (2025)). A model's parametric knowledge affects how well it understands and integrates information during reasoning (reflected in attention patterns), while WM serves as the active processing resource that must load both contextual information and activated parametric knowledge Tighidet et al. (2025). Complex tasks demanding simultaneous consideration of multiple interacting elements can cause cognitive overload regardless of context window utilization or task length. Our pilot study (Appendix B.2) provides proxy evidence that is consistent with this WM framing.

### B.1.3 Detailed Mechanisms of Human Collective Intelligence

Here we elaborate on the cognitive mechanisms enabling human collective intelligence from Section 3.1. Human collective intelligence emerges from sophisticated social-cognitive abilities that enable groups to surpass individual cognitive limitations through several key mechanisms:

**Shared Intentionality and Theory of Mind:** Effective collective intelligence requires individuals to understand others' mental states and coordinate intentions toward common goals (Tomasello et al., 2005; Frith & Frith, 2005), enabling the establishment of common ground necessary for distributed cognitive processing.

**Meta-cognitive Awareness:** Individuals develop meta-knowledge about "who knows what" (Hollingshead, 2001), enabling efficient allocation of cognitive resources and allowing group members to rely on each other for information sharing and retrieval (Hollingshead & Brandon, 2003).

**Spontaneous Organization:** Effective collective intelligence often emerges spontaneously through self-organizing principles (Shteynberg et al., 2023), with groups naturally developing communication patterns and role distributions that optimize cognitive load management.

**Structured Communication:** Groups develop specialized communication protocols that establish common ground, minimizing extraneous cognitive load while maximizing information integration (Tomasello et al., 2005).

These mechanisms demonstrate that human collective intelligence results from emergent properties of social-cognitive interaction that specifically address cognitive load management challenges, providing natural solutions to cognitive overload that inform LLM multi-agent system design.

### B.1.4 Using Cognitive Load Theory to Explain Phenomena in LLM Performance

Here we provide examples of how CLT explains LLM performance phenomena from Section 3.1. Cognitive Load Theory (CLT) offers a valuable lens to interpret puzzling LLM performance issues, positing that LLMs,

like humans, have finite processing capacity. Exceeding this capacity leads to performance degradation. This section concisely analyzes several such cases through CLT.

**Degradation of thought in self-reflection.** Liang et al. (2023) report that LLMs may stick to wrong initial answers under self-reflection. *CLT reading:* holding problem, draft, critique, and revision at once is heavy; if the first pass saturates capacity, the model may superficially agree instead of truly revising—overload.

**Performance drop under dense in-context scaffolding.** Agarwal et al. (2024) find performance can worsen with more in-context examples on hard tasks (e.g., MATH). *CLT reading:* few examples scaffold, but too many raise total load past capacity, like CLT's redundancy effect on working memory.

**Performance vs. NLL under extended in-context examples.** Agarwal et al. (2024) also show scores can fall while NLL improves. *CLT reading:* under overload, models may follow shallow heuristics—fluent text, weak reasoning—because depth is too costly.

**RLHF and reduced diversity.** Kirk et al. (2023) and others note narrower outputs after RLHF despite better instruction following. *CLT reading:* narrow reward shapes impose high conformance load; exploring diverse unrewarded paths costs extra extraneous load, so the model may collapse to a small output shell.

Together, these cases motivate CLT as an analogical lens on LLM limits under heavy informational or processing demand.

## B.2  Pilot Study: Proxy Evidence

This subsection details the pilot study that supplies proxy evidence consistent with the Cognitive Load Theory (CLT) analogy for LLMs, as summarized in Section 3.2. We first define the proxies and motivate them, then describe the protocol and report two controlled experiments, and finally clarify that these measures are diagnostic rather than operational.

### B.2.1  Mathematical Definitions of Cognitive Load Proxies

**Attention Entropy** measures the diversity of the model's attention distribution, formally defined as:

$$H = -\sum_{i=1}^{N} a_i \log a_i \tag{7}$$

where $a_i$ is the normalized attention weight on token $i$ and $N$ is the total number of tokens. Higher entropy indicates more uniform attention distribution across input tokens, suggesting the model must consider more aspects of the input simultaneously.

**Perplexity** measures the model's uncertainty about its predictions, formally defined as:

$$\text{PPL} = \exp\left(-\frac{1}{N}\sum_{i=1}^{N} \log P(w_i)\right) \tag{8}$$

where $P(w_i)$ is the probability assigned to token $w_i$ and $N$ is the sequence length. Lower perplexity indicates higher confidence in predictions.

### B.2.2  Theoretical Justification for Proxies

The two proxies we employ measure **LLM-perceived cognitive load** that must be accommodated within the model's working memory for task completion. This distinction is crucial for understanding what we are measuring. Our analogy starts with the observation that the human brain can only attend to a limited amount of information at once, a feature known as working memory. In cognitive science, working memory capacity refers to how many elements one can simultaneously hold and manipulate; these elements are often referred to as "information chunks." A high cognitive load task requires more working memory as it demands

the simultaneous use of more elements to solve. Therefore, the CL of a task can be reflected by how many "information chunks" are needed to solve it.

**Why Attention Entropy? (Measuring "Information Chunks"):** The architecture of LLMs, due to their inherent attention mechanism, similarly restricts the amount of information they can focus on at once. LLMs predict the next tokens through their attention mechanism, allowing the model to selectively focus on specific, relevant parts of the input context to generate a response. By definition, we can measure how many "information chunks" are being actively considered to complete a task. Thus, using Attention Entropy to measure the sparsity of information integration is a direct proxy. *Low entropy* means sparse, focused attention—few active "elements," lower load. *High entropy* means more uniform attention—many elements at once, higher load.

This analogy is justified through both our experiments and recent theoretical work. In our experiments, we find attention entropy correlates with task complexity (Table 7); also we find adding reasoning effectively reduces attention entropy even with longer context. Our analogy also matches recent work in theoretically explaining why chain of thoughts works (Wen et al., 2024); they also find CoT is creating more sparse attention; in terms of our framework, it is that reasoning is creating cognitive offloading, allowing the model to process fewer chunks during solving the tasks.

**Why Perplexity? (Measuring Processing Fluency):** Perplexity serves as a proxy for processing fluency, representing the cognitive ease of processing information. Metacognitive research establishes that processing fluency (confidence) correlates with task difficulty and can reflect cognitive load faithfully when the subject is not under cognitive overload (Koriat, 2007). In our pilot study, we specifically measure perplexity on prefilled ground-truth answers and validate it. In contrast, during open generation, cognitive overload causes models to lose metacognitive calibration, often resulting in low-perplexity hallucinations where confidence fails to predict performance (Simhi et al., 2025). Both LLM phenomena align with cognitive science findings on metacognitive judgments under varying load conditions.

### B.2.3 Experimental Setup

**Model:** We conducted experiments using Mistral-7B-v0.3, a mid-sized language model that exhibits clear cognitive limitations while maintaining reasonable performance across diverse tasks.

**Prefilled vs. Generated Answers:** We measure perplexity on prefilled ground-truth answers rather than model-generated responses to avoid heuristic confounds: cognitively overloaded models may produce confident but incorrect responses (Agarwal et al., 2024), which would not reflect the true cognitive load of the underlying task.

**Implementation Details:** For Attention Entropy, we aggregate softmax attention weights, then average over heads and layers.

### B.2.4 Experiment 1: Attention Entropy and Task Complexity

**Dataset Construction:** We built a controlled AMPS-Hard arithmetic set with four difficulty levels from simple to multi-step problems, matched input length across levels to isolate complexity, and kept one question format and domain. Different item types are poor to mix for attention entropy (sensitivity to structure and content), so we stay in one domain with graded difficulty.

Table 7: Attention entropy increases with task complexity (consistent with the proxy). Reasoning steps reduce entropy by helping the model focus on key information.

| Task Complexity | Attention Entropy (No Reasoning) | Attention Entropy (With Reasoning) |
|---|---|---|
| Level 1 | 4.442 | 4.439 |
| Level 2 | 4.796 | 4.726 |
| Level 3 | 5.043 | 4.937 |
| Level 4 | 6.101 | 5.920 |

**Analysis.** Attention entropy increases monotonically with task complexity, indicating that harder tasks require the model to consider more information pieces simultaneously, corresponding to higher cognitive load.

Importantly, this finding is **mathematically non-trivial**. Standard information theory predicts that for a probability distribution over $N$ tokens, attention entropy $H = -\sum a_i \log a_i$ should increase approximately as $\log N$ when $N$ grows large. Thus, longer sequences naturally yield higher entropy. However, Table 7 reveals that adding reasoning steps actually *decreases* attention entropy (e.g., Level 3: 5.043→4.937) despite creating longer context. This seemingly paradoxical result demonstrates that reasoning induces *sparse attention patterns*—the model learns to chunk information into coherent reasoning steps, allowing it to focus on fewer elements simultaneously even as total context grows. This finding aligns with recent theoretical work showing Chain-of-Thought creates sparse sequential dependencies that enhance processing efficiency, and is consistent with our working memory analogy: just as humans chunk information to expand effective WM capacity, LLMs achieve similar cognitive offloading through structured reasoning.

### B.2.5 Experiment 2: Perplexity and Instruction Complexity

**Dataset Construction:** We used the FLASK dataset, specifically filtering for problems within the same category to ensure comparability. We selected pairs of problems where one was marked as requiring "expert knowledge" and another without this marking, representing hard and easy tasks respectively within the same domain.

**Instruction complexity levels.** Level 1: no extra instruction. Level 2: "Think step by step." Level 3: "Please think step by step, focusing on factuality and logical reasoning." Level 4: extended reasoning guidance with named strategies. Level 5: full multi-framework instructions.

**Analysis.** For hard tasks, perplexity initially decreases with instruction complexity (Levels 1-3), indicating that structured guidance helps the model focus and reduces cognitive uncertainty. However, perplexity increases again at higher instruction levels (Levels 4-5), suggesting that excessive instruction becomes an additional cognitive burden—classic extraneous load as predicted by CLT.

For easy tasks, perplexity remains consistently low and slightly increases with instruction complexity, indicating that additional guidance provides no benefit and may even introduce unnecessary extraneous load. This pattern matches CLT's **"redundancy effect"**: when a task is within the model's working memory capacity, additional information—even if relevant—can impair performance by consuming limited cognitive resources without providing commensurate benefits. The differentiated effect across difficulty levels is particularly telling. For hard tasks, instructions reduce perplexity from 120.50 to 85.35, indicating they help the model focus and reduce cognitive uncertainty. For easy tasks, the slight increase (3.37→3.45) suggests that processing unnecessary guidance creates extraneous load that outweighs any potential benefit when the model's existing capacity already suffices.

Table 8: Perplexity patterns are consistent with CLT predictions across instruction complexity levels.

| Instruction Complexity | Perplexity (Hard) | Perplexity (Easy) |
|---|---|---|
| Level 1 | 120.50 | 3.37 |
| Level 2 | 88.97 | 3.42 |
| Level 3 | 85.35 | 3.45 |
| Level 4 | 92.48 | 3.46 |
| Level 5 | 100.71 | 3.46 |

### B.2.6 Proxy Scope: Diagnostic Design Constraints, Not Runtime Operations

**Why not test-time use.** Attention entropy and perplexity are not general inference-time meters. We do *not* propose an inference-time algorithm that dynamically detects overload and rewires routing each step. Overloaded models can answer with fluent heuristics that hide true task difficulty (Agarwal et al., 2024), so

self-generated text is a biased load readout. Perplexity on labels needs references absent at test time. Hence PPL-based assistants often misfire: outputs look "easy" yet stay wrong, and label-based PPL is undefined online.

**How we ensured clean experimental signals.** We treat these proxies as *offline diagnostic tools* for theory validation and architecture design. Confound control: hold task domain and type fixed within each study; match input length within each complexity level; score prefilled ground-truth answers instead of free generations to avoid heuristic masking. The proxies therefore support whether CLT-aligned design choices are plausible, rather than steering live inference—*CoThinker* relies on its fixed architectural mechanisms, not online load meters.

### B.3 Post-Hoc Analysis on Main Experiments Using Proxies

We leverage perplexity as a cognitive load proxy to understand how different CoThinker components help weaker models process information from stronger models' outputs. This analysis provides direct evidence that our components reduce cognitive burden during collaboration.

This section (Table 9, Table 11, and Table 10) provides the complete set of PPL ablation studies for component analysis referenced in Section 5.6.

**Analysis.** Table 9: both Thinking Style Orchestrator and TMS cut PPL—less effort to absorb peer text—with TMS giving the largest drop, consistent with stronger scaffolding for specialized content.

Table 10: PPL falls as peer count $k$ grows, with diminishing returns past moderate $k$; very large $k$ can add noise and dilute use of peer answers (Section 4.3).

Table 11: at $k=3$, *similar* peers yield lowest PPL, *random* helps, *diverse* peers raise PPL through integration cost. Similarity trims extraneous load but risks echo chambers; rewiring with $\beta$ reintroduces diversity and intrinsic-load sharing while keeping small-world structure (Section 4.3; Appendix D.5).

Table 9: PPL changes from baseline when adding Style or TMS. Both components reduce PPL; TMS shows the largest relief.

| Model | Base PPL | +Style PPL | +TMS PPL |
|---|---|---|---|
| Mistral-7B | 6.6314 | 5.0422 | 1.5811 |
| Qwen3-8B | 6.5633 | 3.5814 | 1.6876 |

Table 10: PPL vs number of peer answers ($k$). Increasing $k$ consistently lowers PPL with diminishing returns beyond $k \geq 5$.

| $k$ peers | Mistral-7B PPL | Qwen3-8B PPL |
|---|---|---|
| 1 | 4.1516 | 3.3174 |
| 2 | 3.1343 | 3.1022 |
| 3 | 2.8578 | 3.0995 |
| 4 | 2.6123 | 2.5367 |
| 5 | 2.1221 | 2.1429 |
| 6 | 2.0917 | 1.8572 |

## C Small-World Network Properties and Statistical Rigor

### C.1 Small-World Network Analysis

To check that our Communication Moderator creates small-world network properties as predicted by CLT, we analyze the emergent communication networks using the small-world coefficient $\sigma = (C/C_{rand})/(L/L_{rand})$

Table 11: Effect of selection strategy at $k=3$. Similar peers yield the lowest PPL; random also helps; fully diverse peers show higher PPL, consistent with higher integration cost.

| Selection ($k=3$) | Mistral-7B PPL | Qwen3-8B PPL |
|---|---|---|
| Random (3rand) | 2.0967 | 2.1924 |
| Similar (3sim) | 1.8891 | 1.5399 |
| Diverse (3diverse) | 3.0958 | 2.7905 |

(Humphries & Gurney, 2008), where $C$ is clustering coefficient, $L$ is average path length, and the denominators are random graph baselines.

At each round $t$, we construct a directed weighted graph $G^{(t)} = (\mathcal{A}, E^{(t)}, W^{(t)})$ where edge weights $w_{uv} = 1 - \text{sim}(x_u, x_v)$ represent cognitive distance. We calculate clustering coefficient, path length, and compare against 100 random graphs with the same degree distribution.

All configurations yield $\sigma > 1$, consistent with small-world properties. For M=6, N=3, $\beta$=0.3: median $\sigma$=2.75; M=12: $\sigma$=2.87; M=18: $\sigma$=3.12. This pattern is consistent with the Communication Moderator producing high local clustering (efficient refinement of similar ideas) with short average paths (rapid global propagation of diverse insights), matching the qualitative structure CLT motivates.

## C.2 Statistical Significance Analysis

**Bootstrap Standard Error and Paired-Test Methodology.** We use bootstrap resampling to estimate the standard error (SE) of our performance metrics across test instances. We use greedy decoding (temperature=0) for all refinement rounds to ensure deterministic behavior. Initial generation uses temperature=0.25 to create diverse agent starting points, but subsequent refinement is deterministic. Multiple random seeds would yield highly similar results under greedy decoding. Therefore, we report bootstrap variance representing instance-wise variance under large samples rather than seed variance.

For each task and model we pool instance scores, draw $B$=2000 bootstrap resamples with replacement, and set SE to the standard deviation of the bootstrap means. For significance claims, we use paired tests on per-question outcomes (*CoThinker* vs. each baseline) and apply Holm–Bonferroni correction within the predefined test families. This combination is appropriate because paired tests exploit within-question matching to reduce noise, while Holm–Bonferroni controls family-wise false positives across the multiple baseline comparisons without being as conservative as plain Bonferroni. Concretely, for baseline $b$ and question $q \in \{1, \ldots, n\}$ we define $d_q^{(b)} = s_q^{\text{Ours}} - s_q^{(b)}$ and test $H_0 : \mathbb{E}[d_q^{(b)}] = 0$ with a paired test; if the resulting family of $m$ $p$-values is ordered as $p_{(1)} \leq \cdots \leq p_{(m)}$, Holm rejects $H_{(k)}$ while $p_{(k)} \leq \alpha/(m - k + 1)$.

**Bootstrap Standard Error Results.**

Table 12 shows complete bootstrap SE results for our main comparison. The small sampling variance across all methods and model families demonstrates high reliability and stability of our bootstrap estimates.

**High-load task significance.** On Reasoning (high intrinsic load), the *CoThinker* vs. IO differences are evaluated using paired tests on per-question outcomes; significance is determined from the resulting paired-test $p$-values after Holm correction, consistent with the main-text protocol.

**Per-cell uncertainty vs. significance testing.** The ratio is computed per subtask and averaged within a category, so it does not equal the aggregate score divided by the Flash-8B IO aggregate when subtask anchor accuracies differ across the category. The bracket below each cell in Table 2 reports bootstrap SE on each method's mean ($B$=2,000, within-subtask resampling). These SE values are descriptive uncertainty for each cell. Significance claims are determined separately by paired tests on per-question outcomes with Holm correction, as reported in Table 13.

**Decoding stability and interpretation of bootstrap SE.** Refinement rounds use greedy decoding (temperature 0); only initial generation uses temperature 0.25. Reported bootstrap SE therefore reflects

Table 12: Bootstrap Standard Error (SE) for all methods across model families, rescaled by 8B IO baseline. Values shown are rescaled SE (original SE divided by 8B IO baseline mean for each task). The small sampling variance (rescaled SE ranging from 0.034 to 0.318) demonstrates high reliability and stability of our bootstrap estimates.

| Model | Method | Math | Data | Reasoning | Language | Inst. Follow. | Overall |
|---|---|---|---|---|---|---|---|
| Gemini-Flash-8B | IO | 0.139 | 0.146 | 0.180 | 0.177 | 0.038 | 0.043 |
| | CoT | 0.146 | 0.149 | 0.185 | 0.178 | 0.037 | 0.042 |
| | SR | 0.145 | 0.116 | 0.164 | 0.171 | 0.043 | 0.041 |
| | MAD | 0.144 | 0.146 | 0.186 | 0.152 | 0.043 | 0.043 |
| | DMAD | 0.138 | 0.140 | 0.189 | 0.174 | 0.043 | 0.042 |
| | CoThinker | 0.156 | 0.140 | 0.209 | 0.288 | 0.040 | 0.045 |
| Gemini-Flash | IO | 0.172 | 0.148 | 0.209 | 0.213 | 0.036 | 0.044 |
| | CoT | 0.167 | 0.143 | 0.211 | 0.193 | 0.036 | 0.045 |
| | SR | 0.158 | 0.103 | 0.204 | 0.207 | 0.043 | 0.045 |
| | MAD | 0.165 | 0.136 | 0.215 | 0.204 | 0.041 | 0.045 |
| | DMAD | 0.169 | 0.141 | 0.216 | 0.204 | 0.037 | 0.044 |
| | CoThinker | 0.191 | 0.147 | 0.231 | 0.311 | 0.038 | 0.045 |
| Gemini-Pro | IO | 0.264 | 0.136 | 0.240 | 0.311 | 0.034 | 0.045 |
| | CoT | 0.253 | 0.133 | 0.233 | 0.283 | 0.035 | 0.046 |
| | SR | 0.266 | 0.116 | 0.233 | 0.290 | 0.036 | 0.044 |
| | MAD | 0.241 | 0.127 | 0.249 | 0.261 | 0.038 | 0.046 |
| | DMAD | 0.243 | 0.141 | 0.239 | 0.285 | 0.036 | 0.046 |
| | CoThinker | 0.225 | 0.133 | 0.244 | 0.318 | 0.038 | 0.044 |

*question-sampling* variance ("what if the test set were a different 100 questions?"), not *seed* variance. With nearly deterministic refinement, per-cell point estimates are reproducible run-to-run, and the directional ranking of methods is stable across bootstrap settings.

**Holm-corrected significance.** Table 13 reports per-question paired-bootstrap tests ($B$=2,000) aggregated across the three Gemini models per category. The effect size is $\Delta_{\mathrm{norm}} = \frac{1}{|S|} \sum_s (\overline{\mathrm{acc}}_{\mathrm{Ours},s} - \overline{\mathrm{acc}}_{b,s})/\overline{\mathrm{acc}}_{\mathrm{8B\text{-}IO},s}$; $\Delta_{\mathrm{pp}}$ is the raw mean gap. Holm–Bonferroni is applied within two families that map to distinct claims: vs. single-agent (IO/CoT/SR; 15 tests) tests *whether multi-agent helps at all*; vs. multi-agent (MAD/DMAD; 10 tests) tests *whether the CLT design helps over heuristic multi-agent.* Sixteen of 25 comparisons are Holm-significant: 13 wins + 2 losses on single-agent; 3 wins + 1 loss on multi-agent. All 3 losses fall on instruction-following, the CLT-predicted low-intrinsic-load boundary.

# D System Design and Implementation Details

## D.1 CoThinker Architecture Details

### D.1.1 CoThinker as Plug-and-Play CLT Principles

CoThinker packages CLT-derived **cognitive design principles** as "plug-and-play" add-ons to engineering-first stacks (Section 4). Examples: AutoGen (Wu et al., 2023), CrewAI, MetaGPT (Hong et al., 2023), ChatDev (Qian et al., 2023), AgentVerse (Chen et al., 2023b), LangGraph, Multi-Agent Debate (Du et al., 2023), ReConcile (Chen et al., 2023a). Those systems emphasize coordination; we add load management (styles, moderated references, TMS) for layering (e.g., load-aware chat, TMS in long runs).

## D.2 Information Flow and Message Passing

Per agent, round $t$ (Figure 2): **(i)** moderator picks $N$ peers; TMS injects compressed state (consensus, expertise). **(ii)** Agent merges peer text and TMS (offloading work peers already cover). **(iii)** Updated draft;

TMS refresh; moderator rebuilds the reference graph. TMS = low-bandwidth summary; moderator channel = peer payloads. Each round: in-degree $N$, similarity ties, $\beta$-rewiring vs. broadcast/static graphs.

Table 13: Per-category Holm-corrected paired-bootstrap tests of *CoThinker* against each baseline. $\Delta_{\mathrm{norm}}$ is the per-subtask Flash-8B-IO-normalized effect size; $\Delta_{\mathrm{pp}}$ is the raw accuracy gap in percentage points. $p$-values from $B$=2000 paired bootstrap. *Family A*: Holm–Bonferroni over the 15 single-agent tests. *Family B*: Holm–Bonferroni over the 10 multi-agent tests. ✓ = Holm-significant under the relevant family at $\alpha$=0.05; ✗ = Holm-significant loss.

| Category | Baseline | $\Delta_{\mathrm{norm}}$ | $\Delta_{\mathrm{pp}}$ | $p$ | Holm |
|---|---|---|---|---|---|
| Math | vs. IO | +16.2% | +5.3 | $<.001$ | A: ✓ |
| | vs. CoT | +15.5% | +5.1 | 0.001 | A: ✓ |
| | vs. SR | +18.3% | +6.0 | $<.001$ | A: ✓ |
| | vs. MAD | +2.3% | +0.8 | 0.63 | — |
| | vs. DMAD | +6.3% | +2.2 | 0.16 | — |
| Data Analysis | vs. IO | +31.2% | −1.2 | 0.006 | A: ✓ |
| | vs. CoT | +39.0% | −0.8 | 0.009 | A: ✓ |
| | vs. SR | +123.6% | +26.3 | $<.001$ | A: ✓ |
| | vs. MAD | +46.4% | +12.8 | $<.001$ | B: ✓ |
| | vs. DMAD | +24.9% | +1.6 | 0.04 | — |
| Reasoning | vs. IO | +19.4% | +3.8 | 0.021 | A: ✓ |
| | vs. CoT | +20.7% | +3.9 | 0.015 | A: ✓ |
| | vs. SR | +41.6% | +10.0 | $<.001$ | A: ✓ |
| | vs. MAD | +11.2% | +2.2 | 0.21 | — |
| | vs. DMAD | +19.7% | +4.5 | 0.02 | — |
| Language | vs. IO | +75.6% | +12.2 | $<.001$ | A: ✓ |
| | vs. CoT | +80.3% | +12.9 | $<.001$ | A: ✓ |
| | vs. SR | +67.9% | +10.9 | $<.001$ | A: ✓ |
| | vs. MAD | +47.7% | +7.7 | $<.001$ | B: ✓ |
| | vs. DMAD | +15.2% | +2.4 | 0.17 | — |
| Instr. Follow. | vs. IO | −6.9% | −5.8 | $<.001$ | A: ✗ |
| | vs. CoT | −5.9% | −5.0 | $<.001$ | A: ✗ |
| | vs. SR | +17.8% | +15.0 | $<.001$ | A: ✓ |
| | vs. MAD | +9.7% | +8.2 | $<.001$ | B: ✓ |
| | vs. DMAD | −3.8% | −3.2 | 0.002 | B: ✗ |

### D.3 Prompt Architecture for Agent Parallel Thinking

Parallel Thinking varies *styles* (Sternberg (Sternberg, 1997))—ways of deploying ability, not new skills—avoiding heavy personas; in-context learning shifts style more than core knowledge (Lin et al., 2024; Zhao et al., 2025). Two-stage prompts:

**1. Style Orchestration (**Orch**).** An LLM maps task $D$ and Sternberg dimensions (Functions: Legislative, Executive, Judicial; Forms; Levels Global/Local; Scope Internal/External; Leanings) to $\{\phi_1, \ldots, \phi_M\}$ for $\{A_i\}$, adapting each $\psi_i$ to $D$.

Orchestrator template (e.g. $\psi_i$ = "Legislative–Global"):

```
Given the primary task: "{Task D}"
And the base thinking style profile (from Sternberg's Theory of
Mental Self-Government): "{Base Style profile psi_i, e.g.,
Legislative function with a Global level preference}"

Generate a concise (1-2 sentences) task-specific adaptation
```

```
of this thinking style profile that would be most beneficial
for an agent contributing to this primary task. The agent
should focus its reasoning and output according to this
adapted style.
Task-Specific Style for an agent:
```

**2. Agent instruction** (Agent). Each $A_i$ carries $\phi_i$ for the full trajectory. Template:

```
You are Agent {num}. Your assigned thinking style for this
task is: "{Style phi_i generated by Orchestrator}".
The overall task is: "{Task D}".
[Other contextual information, e.g., from TMS mu^(t),
references P_i^(t-1), own previous thought x_i^(t-1)]

Keeping your assigned thinking style in mind, please provide
your thoughts/solution:
```

### D.4 Prompt Architecture for Transactive Memory System (TMS) Emulation

The TMS manager acts as an *information bottleneck*: raw peer outputs and prior-round chatter are distilled into a structured collective state $\mu^{(t)}$ with fixed semantic slots, so downstream agent prompts carry high-signal, lower-token summaries instead of unbounded chat history—the engineering counterpart of context compression under CLT.

Section 4.2: we emulate human TMS—who knows what, access/integrate distributed knowledge, trust (Wegner, 1987; Hollingshead, 2001; Lewis, 2003); *encoding, storage, retrieval* via *specialization, credibility, coordination* (Yoo & Kanawattanachai, 2001).

Each round $t$, a TMS Manager LLM fills a template from $\{x_j^{(t-1)}\}_{j=1}^M$ and $\mu^{(t-1)}$ into structured $\mu^{(t)}$. Required slots:

**Expertise directory.** Per-agent highlights from round $t-1$, tied to $\phi_j$ or roles when helpful, e.g. *"Agent A (Analytical) flagged three data inconsistencies; Agent B (Creative) proposed two solutions on X."* Supports *encoding* and *retrieval* cues.

**Shared store (consensus / artifacts).** Emerging agreement, facts, partial solutions, e.g. *"Consensus: bottleneck is allocation. Established: budget $\leq Y$."* Validated *storage* so agents need not re-derive.

**Divergences / open issues.** Conflicts, unanswered points, e.g. *"C favors Alpha; D favors Beta. Open: feasibility of X by deadline."* Steers *coordination* next round.

Agent prompt at round $t$ (excerpt):

```
[Agent's assigned thinking style: {Style_phi_i}]
[Overall Task: {Task_D}]

Collective Summary from Previous Round (reflecting shared understanding mu^(t)):
"{Text of mu^(t) generated by the TMS Manager using the TMS Template}"

Your Previous Output (x_i^(t-1)):
"{Text of x_i^(t-1)}"

Reference Outputs from Peers (P_i^(t-1)):
Reference 1 (from Agent A_k): "{Text of x_k^(t-1)}"
Reference 2 (from Agent A_l): "{Text of x_l^(t-1)}"
...
```

```
Based on all the above, and keeping your thinking style in mind,
provide your refined thoughts/contribution for the current round:
```

$\mu^{(t)}$ thus encodes directory, consensus, and open issues—structured shared memory, not ad hoc chat.

### D.5 Communication Moderator: Cultivating an Efficient Network via Strong and Weak Ties

Communication cost matters in Collaborative CLT (Kirschner et al., 2009; 2018). The moderator (Section 4.3) sets ties using network ideas to trade load vs. group performance.

**Local cohesion (strong ties, clustering).** With probability $1-\beta$, $A_i$ links to peers with outputs most similar to $x_i^{(t-1)}$—**strong ties** (Granovetter, 1983), high **local clustering**. Similarity: cosine on embeddings (e.g. `all-MiniLM-L6-v2`). Supports local refinement and cuts extraneous duplication.

**Global integration (weak ties, small-world).** $\beta=0$ risks echo chambers and long paths—**structural holes** (Burt, 2004), poor global mixing. With probability $\beta$, random peers add **weak ties** (Granovetter, 1983) and **small-world** structure (Watts & Strogatz, 1998). Here, $\beta$ spreads diverse views, shortens paths, shares intrinsic load, and slows premature lock-in—bridging local cohesion and global reach (Section 4.3).

### D.6 Synthesizer Module: Consolidation and Cognitive Grounding

The Synthesizer (Section 4.4) maps $\{x_i^{(T-1)}\}_{i=1}^M$ and $\mu^{(T-1)}$ to one answer after $T_{\max}$ rounds—either an **external** dedicated LLM or an **in-group** agent (Lu et al., 2024; Shinn et al., 2023). **External:** reads all finals $+ \mu^{(T-1)}$; akin to **observational learning** (Bandura & Walters, 1977) (integrate without prior round load). **In-group:** one collaborator synthesizes from context $+ \mu^{(T-1)} +$ peers; aligns with CCLT (Kirschner et al., 2018) shared regulation / "collaborative leading."

**External Synthesizer (**Synth**) template:**

```
Original Task:  "[Task Description D]"
After collaborative thinking, the final individual perspectives from M=[Number of Agents]
agents are:
Agent 1:  "[x₁^(T-1)]"
...
Agent M: "[x_M^(T-1)]"
The final collective understanding synthesized during their collaboration is:
"[μ^(T-1)]"
Based on all this information, please generate a comprehensive, high-quality, and coherent
final solution to the original task.
```

## E   Supplementary Experimental Data and Benchmark Details

### E.1   Experimental Configuration

Configuration matches Section 5.1.

**LLM API parameters.** Temperature 0.25 for IO, CoT, SR and for round $t=0$ of MAD, DMAD, and *CoThinker*; for $t>0$, temperature 0.0 and `frequency_penalty = 0.5` for multi-agent iterative methods. Other API defaults (`top_p`, `top_k`, etc.) unchanged. Max output tokens set per task as needed.

***CoThinker* defaults.** Unless an ablation states otherwise: $M=6$, $T_{\max}=3$ (one initial round plus two refinements), $N=3$ peer references per agent, $\beta=0.3$.

**Baselines.**   We compare against the following implementations:

- **Single Agent (IO).** Task instruction only, no specialized prompting (raw capability baseline).

- **Single Agent (CoT).** Chain-of-thought prompting (Wei et al., 2022): the model reasons step by step or sees few-shot reasoning demos before the final answer.
- **Single Agent (Self-Refine; SR) (Madaan et al., 2023).** Three iterations: generate, critique, revise.
- **Multi-Agent Debate (MAD) (Liang et al., 2023; Du et al., 2023).** $M=6$, $T=3$; each round every agent sees all peers' prior solutions, may critique, and refines; final answer from the best-performing agent after debate.
- **Diverse Multi-Agent Debate (DMAD) (Liu et al., 2025b).** As MAD, but distinct first-round strategies per agent (IO, CoT, Step-Back, etc.) before shared refinement.

## E.2 Benchmark Details

**LiveBench.** LiveBench (White et al., 2025) pools difficult, objectively scored tasks (e.g., competition math, logic and word puzzles) drawn in part from Big-Bench Hard (Suzgun et al., 2023), AMPS (Hendrycks et al., 2021), and IFEval (Zhou et al., 2023), with periodic refreshes to limit contamination. Domains span mathematics, reasoning, language, instruction following, and data analysis. Difficulty targets high cognitive load even for strong models.

**CommonGen-Hard.** From CommonGen (Lin et al., 2020) via Madaan et al. (2023): generate a coherent multi-sentence paragraph using 3–5 target concepts amid ∼30 distractors. We score with the ten-dimensional rubric of Li et al. (2018) via an LLM judge (Gemini-1.5-Pro) under the Ye et al. (2023) protocol, identical to Section 5.3 (Likert 1–10 per dimension, then aggregate). This stresses element interactivity while keeping evaluation systematic.

## E.3 Cross-Model Detailed Results

This section provides comprehensive cross-model evaluation results referenced in Section 5.4, including detailed performance breakdowns across multiple LLM families and task categories.

Table 14 presents the full cross-model evaluation results across all tested models and evaluation scenarios. Table 15 provides detailed task-specific results for the standard setting cross-model evaluation, showing CoThinker's performance across individual reasoning and mathematical tasks.

Table 14: Complete cross-model evaluation results on LiveBench under constrained setting (8192 tokens, greedy decoding). Values in parentheses are Bootstrap Standard Errors (SE) in percentage points. CoThinker shows directional improvements across diverse model families and capabilities; per-cell significance not corrected here.

| Model | Method | Avg (SE) | Math (SE) | Reasoning (SE) |
|---|---|---|---|---|
| Gemini-2.5-Flash | CoThinker | **72.8 (2.67)** | **76.3 (2.57)** | **69.2 (4.15)** |
| | DMAD | 59.7 (3.32) | 56.7 (4.48) | 62.8 (4.70) |
| | IO (Baseline) | 45.1 (3.40) | 59.3 (3.78) | 31.0 (4.86) |
| GPT-4.1-Mini | CoThinker | **55.4 (2.92)** | **40.0 (3.72)** | **70.8 (3.90)** |
| | DMAD | 39.1 (3.21) | 34.2 (3.84) | 44.0 (4.82) |
| | IO (Baseline) | 37.4 (3.27) | 34.0 (3.91) | 40.8 (4.82) |
| Qwen3-32B | CoThinker | **22.1 (2.89)** | **18.9 (3.52)** | **25.2 (4.22)** |
| | DMAD | 11.5 (2.38) | 8.8 (3.53) | 14.2 (3.25) |
| | IO (Baseline) | 11.7 (2.33) | 3.4 (3.53) | 20.0 (3.14) |
| DeepSeek-R1-8B | CoThinker | **5.8 (2.28)** | 2.9 (1.42) | **8.8 (3.58)** |
| | DMAD | 5.2 (1.71) | **3.8 (2.00)** | 6.5 (2.58) |
| | IO (Baseline) | 2.3 (1.46) | 1.9 (1.94) | 2.8 (2.00) |

Table 15: Task-specific performance breakdown across models and tasks. Full CoThinker configuration outperforms the baseline single-agent approach on most reasoning and mathematical tasks, consistent with the complete multi-agent architecture providing the load-distribution effects targeted by the design.

| Model Configuration | Zebra Puzzle | Spatial | Math Comp |
|---|---|---|---|
| GPT-5-Nano (CoThinker) | 75.75 | 88.00 | 89.13 |
| GPT-5-Nano (IO) | 56.75 | 80.00 | 78.26 |
| Qwen3-30B-A3B (CoThinker) | 83.50 | 84.00 | 84.78 |
| Qwen3-30B-A3B (IO) | 79.00 | 76.00 | 73.91 |
| GPT-OSS-20B (CoThinker) | 55.00 | 80.00 | 78.26 |
| GPT-OSS-20B (IO) | 54.00 | 78.00 | 71.74 |

## E.4 Other Component Ablation Details

This section provides the complete component ablation results referenced in Section 5.6, showing detailed performance breakdowns for other model configurations tested. All configurations use M=6 agents and N=3 references with greedy decoding.

Table 16: TMS ablation for Gemini-1.5-Flash-8B.

| Task | TMS: ON | TMS: OFF |
|---|---|---|
| Math: Math Comp | 35.6 | 7.7 |

Table 17: TMS ablation for Gemini-1.5-Flash.

| Task | TMS: ON | TMS: OFF |
|---|---|---|
| Math: Olympiad | 47.1 | 44.2 |

Table 18: TMS ablation for Gemini-1.5-Pro.

| Task | TMS: ON | TMS: OFF |
|---|---|---|
| Data Analysis: CTA | 58.0 | 58.0 |
| Instruction Following: Paraphrase | 69.8 | 70.2 |
| Instruction Following: Simplify | 68.0 | 68.9 |
| Instruction Following: Summarize | 68.4 | 66.8 |
| Language: Connections | 53.5 | 52.6 |
| Reasoning: Zebra Puzzle | 48.2 | 39.5 |

## E.5 Task-Wise Performance Data

*This subsection provides comprehensive raw scores for all subtasks across various model families and prompting methodologies, along with ablation studies investigating sensitivity to key hyperparameters.*

## E.6 Raw Subtask Performance Scores

The subsequent tables (Table 19 through Table 21) itemize the raw performance scores achieved on each subtask. Scores are reported to two decimal places. A hyphen (-) signifies missing or non-numeric data. Each table is dedicated to a distinct base model family.

Table 19: Raw scores for each subtask for Gemini-1.5-Flash-8B models across different prompting methods.

| Subtask | IO | CoT | SR | MAD | DMAD | CoThinker |
|---|---|---|---|---|---|---|
| Connections | 13.50 | 18.17 | 17.33 | 17.67 | 17.00 | 19.33 |
| CTA | 54.00 | 50.00 | 30.00 | 48.00 | 52.00 | 54.00 |
| Math Comp. | 26.09 | 23.91 | 21.74 | 28.26 | 30.43 | 26.09 |
| Olympiad | 23.82 | 27.64 | 23.84 | 28.25 | 25.87 | 29.00 |
| Paraphrase | 74.27 | 72.82 | 38.42 | 65.22 | 66.55 | 46.02 |
| Simplify | 70.33 | 70.70 | 62.78 | 63.88 | 61.08 | 70.25 |
| Spatial | 34.00 | 28.00 | 18.00 | 34.00 | 22.00 | 28.00 |
| Story Gen. | 73.08 | 68.75 | 62.92 | 66.75 | 67.00 | 65.08 |
| Summarize | 69.35 | 71.27 | 50.43 | 58.32 | 62.62 | 42.32 |
| Table Join | 5.44 | 4.10 | 0.00 | 2.00 | 1.78 | 12.02 |
| Table Reformat | 80.00 | 82.00 | 36.00 | 38.00 | 50.00 | 60.00 |
| Zebra Puzzle | 16.00 | 22.25 | 17.25 | 22.75 | 17.00 | 25.75 |

Table 20: Raw scores for each subtask for Gemini-1.5-Flash models across different prompting methods.

| Subtask | IO | CoT | SR | MAD | DMAD | CoThinker |
|---|---|---|---|---|---|---|
| Connections | 28.17 | 24.00 | 22.83 | 33.17 | 28.50 | 33.67 |
| CTA | 56.00 | 56.00 | 36.00 | 56.00 | 54.00 | 52.00 |
| Math Comp. | 41.30 | 39.13 | 39.13 | 41.30 | 41.30 | 41.30 |
| Olympiad | 32.20 | 34.37 | 33.35 | 34.41 | 33.27 | 36.89 |
| Paraphrase | 80.70 | 78.17 | 52.22 | 80.58 | 82.22 | 72.35 |
| Simplify | 75.83 | 77.68 | 67.57 | 72.07 | 74.40 | 69.00 |
| Spatial | 50.00 | 50.00 | 36.00 | 58.00 | 52.00 | 52.00 |
| Story Gen. | 76.25 | 77.50 | 57.92 | 60.75 | 80.75 | 79.50 |
| Summarize | 77.55 | 75.92 | 54.05 | 68.47 | 74.33 | 68.97 |
| Table Join | 21.64 | 22.78 | 8.12 | 15.00 | 32.60 | 31.20 |
| Table Reformat | 86.00 | 80.00 | 44.00 | 48.00 | 44.00 | 50.00 |
| Zebra Puzzle | 28.50 | 32.00 | 32.50 | 34.25 | 37.50 | 38.50 |

Table 21: Raw scores for each subtask for Gemini-1.5-Pro models across different prompting methods.

| Subtask | IO | CoT | SR | MAD | DMAD | CoThinker |
|---|---|---|---|---|---|---|
| Connections | 31.17 | 36.50 | 35.17 | 44.67 | 44.50 | 46.00 |
| CTA | 56.00 | 58.00 | 36.00 | 56.00 | 60.00 | 58.00 |
| Math Comp. | 47.83 | 36.96 | 45.65 | 54.35 | 56.52 | 56.52 |
| Olympiad | 51.79 | 54.77 | 50.16 | 59.63 | 58.46 | 62.72 |
| Paraphrase | 75.37 | 73.78 | 34.18 | 48.50 | 73.88 | 65.17 |
| Simplify | 74.77 | 75.72 | 54.48 | 55.43 | 72.88 | 66.37 |
| Spatial | 44.00 | 48.00 | 36.00 | 34.00 | 38.00 | 38.00 |
| Story Gen. | 69.72 | 68.05 | 42.55 | 56.85 | 67.30 | 73.05 |
| Summarize | 68.92 | 67.17 | 46.23 | 52.83 | 69.05 | 65.72 |
| Table Join | 35.98 | 32.56 | 16.16 | 43.82 | 42.32 | 44.18 |
| Table Reformat | 88.00 | 88.00 | 28.00 | 28.00 | 86.00 | 78.00 |
| Zebra Puzzle | 39.00 | 35.75 | 40.75 | 41.00 | 42.25 | 44.50 |

### E.7 Subtask Descriptions

The evaluation benchmark comprises a diverse array of subtasks, each designed to assess specific reasoning and generation capabilities of the models. Concise descriptions for each subtask category are provided below:

**Connections**: Assesses the model's aptitude for identifying and comprehending relationships (e.g., logical, causal, shared attributes) between disparate textual elements or conceptual ideas.
**CTA (Call to Action)**: Evaluates the model's effectiveness in generating or interpreting persuasive or directive language aimed at eliciting a targeted response or action.
**Math Comp. (Mathematical Computation)**: Measures the model's proficiency in executing mathematical calculations and resolving problems necessitating computational procedures.
**Olympiad**: Challenges the model with highly complex mathematical problems, characteristic of mathematics Olympiads, which demand profound reasoning and multi-step solution strategies.
**Paraphrase**: Tests the model's ability to accurately rephrase given text while preserving its original semantic content, thereby demonstrating linguistic understanding and versatility.
**Simplify**: Assesses the model's capacity to transform complex textual information into a more readily understandable format, typically by employing simpler vocabulary and sentence structures without loss of core meaning.
**Spatial**: Evaluates the model's spatial reasoning faculties, including its ability to understand and reason about objects in two or three-dimensional space, their interrelations, positions, and transformations.
**Story Generation**: Measures the model's creative ability to produce coherent, engaging, and contextually relevant narratives derived from specified prompts or constraints.
**Summarize**: Assesses the model's proficiency in condensing extended passages of text into succinct summaries that encapsulate the principal points and essential information.
**Table Join**: Evaluates the model's comprehension of relational data structures by requiring it to identify appropriate mechanisms for combining or linking multiple data tables based on common columns or keys.
**Table Reformat**: Tests the model's capability to manipulate tabular data by converting a table from one structural or data representation format to another, adhering to provided instructions.
**Zebra Puzzle**: Assesses the model's deductive reasoning and constraint satisfaction abilities through logic puzzles (such as Einstein's Puzzle) that necessitate deriving a solution from a given set of clues.

### E.8 Ablation Studies

All ablations use Gemini-1.5-Flash-8B. Hyperparameter sweeps and corresponding figures:

| Study | Swept / Config | Figure |
|---|---|---|
| Reference set size (N) | $N \in \{0, \dots, 5\}$; peer messages per agent | 5a |
| Exploration rate ($\beta$) | $\beta$ varied; $N$=2 fixed | 5b |
| Number of agents (M) | $M$ varied; $N$=3 fixed | 6 |
| M/N configurations | M6_N3, M12_N6, M18_N3 (with style) | 7 |

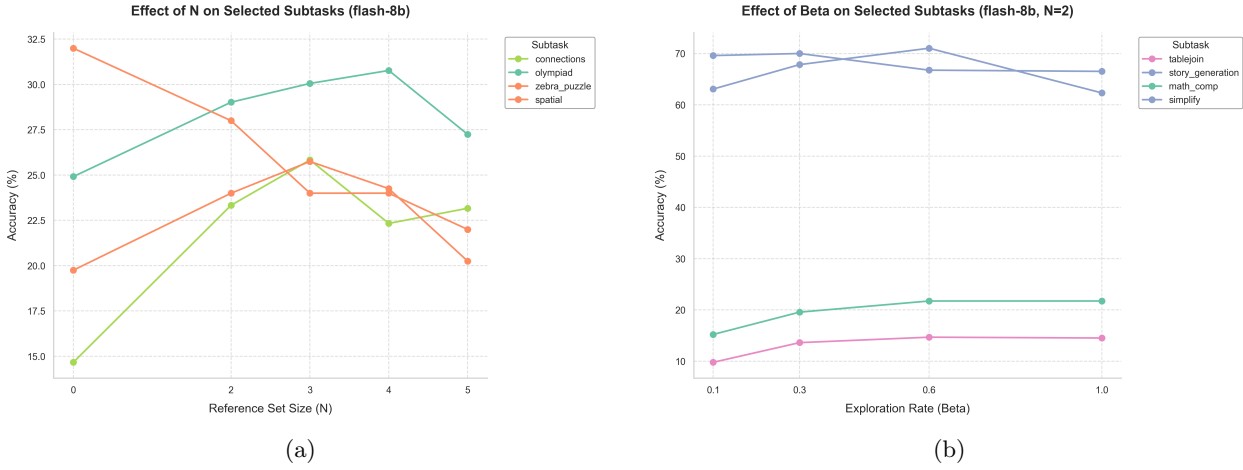

(a)                                                                                    (b)

Figure 5: Ablation studies on Gemini-1.5-Flash-8B. (a) Effect of Reference Set Size (N) on 'connections', 'olympiad', 'zebra_puzzle', 'spatial'. (b) Effect of Exploration Rate (Beta), N=2, on 'tablejoin', 'story_generation', 'math_comp', 'simplify'. Subtasks are color-coded by category.

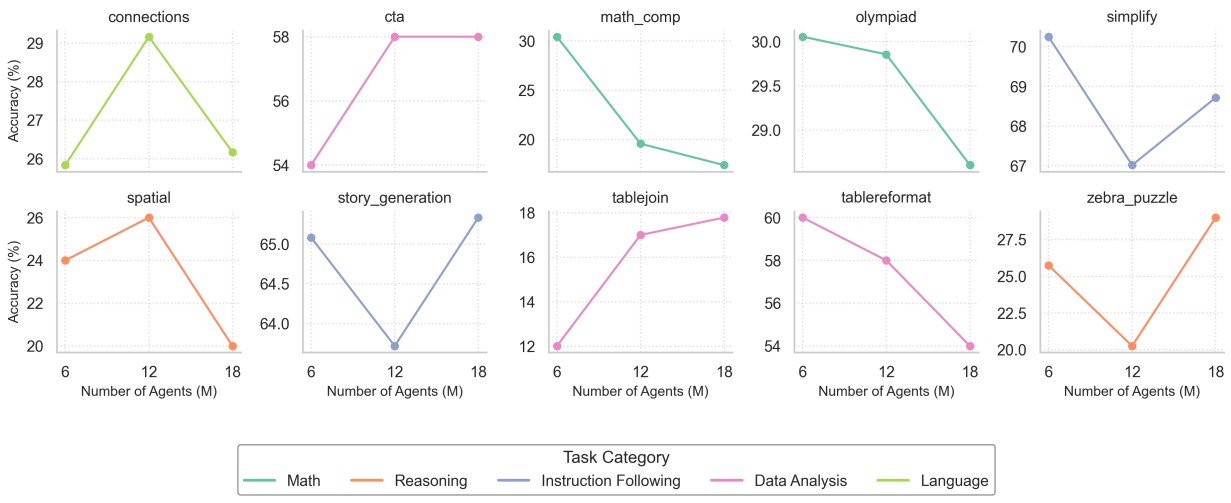

Figure 6: Effect of Number of Agents (M) on performance across all subtasks for Gemini-1.5-Flash-8B with N=3. Each facet corresponds to a subtask, color-coded by its primary category.

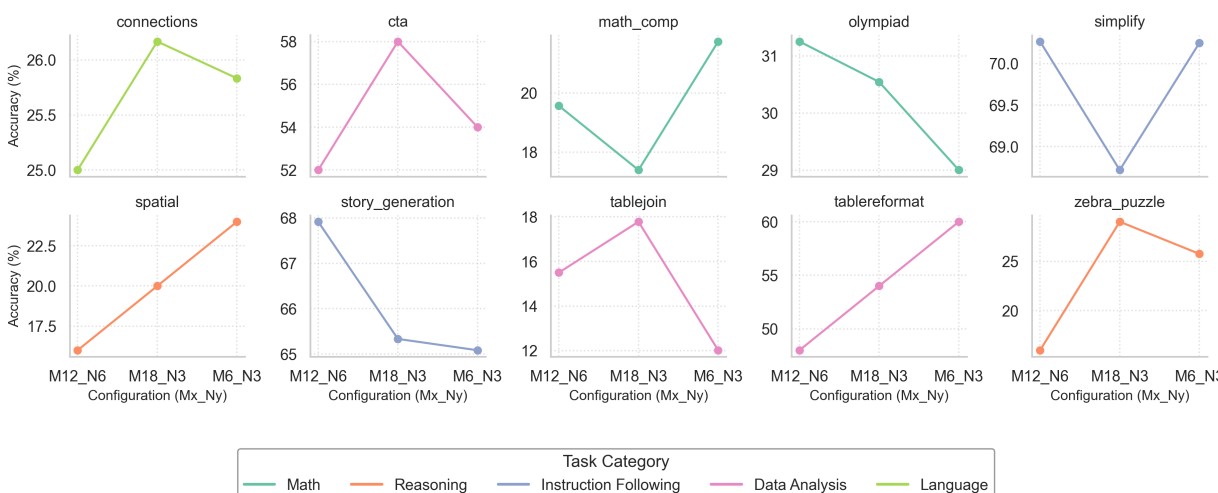

Figure 7: Subtask performance for specific M/N configurations (M6_N3, M12_N6, M18_N3) using Gemini-1.5-Flash-8B under the configuration (Beta=0.3, T=3). Faceted by subtask.

### E.9 Inference-cost estimation protocol

Table 6 reports per-question output tokens on the high-intrinsic-load LiveBench subset; both the *CoThinker* and baseline numbers come from the canonical configurations used to populate Table 2. To keep the comparison apples-to-apples we uniformly identify and drop two per-question failure modes: *(i)* agent traces stop generating coherently and loop on a short repeated span, which inflates token count without contributing to the answer; and *(ii)* baseline runs whose output is empty or near-empty, indicating refusal rather than a real attempt. The same drop list is then applied to every method before we take the per-task median over the surviving questions and average across tasks.

## F  Extended Discussion

### F.1  Ethics and reproducibility

**Ethics and societal impact.**   This study advances basic understanding of cognitive load in LLM collaboration and does not involve human subjects or collection of personal data; evaluations use public benchmarks and commercial LLM APIs under their terms of service. More capable multi-agent coordination could still accelerate low-quality or misleading content generation, increase competitive pressure in knowledge work if deployed without safeguards, or concentrate influence where only well-resourced actors can run large agent pools. A specific concern is that load-managed multi-agent systems are designed to compose information across many sources without exceeding any single agent's attention budget; the same capability can be turned toward constructing tailored persuasive content that draws on a target audience's prior beliefs, demographic context, and topical priors at once, with different agents specialising in factual framing, emotional appeal, and social proof. Because the moderator keeps each agent's local load tractable, such systems can sustain coherent, multi-step persuasion over longer interaction horizons than a single agent reliably can. The risk is therefore not primarily disinformation but the cheap automation of strategically adaptive persuasion at advertising or political-campaign volumes. We scope claims to controlled benchmarks, foreground limitations, and encourage deployment-time guardrails—including human oversight on persuasion-sensitive outputs, rate limits on high-throughput synthetic-content pipelines, and domain-specific risk review—for any adaptation beyond research settings.

**Reproducibility.**  Reproduction relies on standard APIs and public benchmarks.  The main text and Appendix D specify models, prompts, communication protocols, and workflow; Appendix E records defaults, baselines, and supplementary tables. LiveBench and CommonGen-Hard use published task definitions and scoring conventions as cited in the main paper.

### F.2  Limitations and future directions

Our work explores the utility of Cognitive Load Theory (CLT) as a generative design lens for multi-agent LLM systems.  The limitations of this initial study are best understood as defining the current boundaries of our methods and charting a course for developing more robust tools for this new area of research.

**Toward runtime cognitive load measurement and adaptive systems.**  A central challenge is better **runtime measurement** of cognitive state in LLMs, and identifying operational cognitive-load metrics that work at inference time without ground truth is a major future direction for this work.  Our proxies need ground truth, so they suit design-time checks more than live adaptation. Adaptive overload detection likely needs progress on three fronts:

- **Chunking and working memory.** How LLMs group tokens into representational "elements" beyond raw length.
- **Load along a reasoning chain.** How demand shifts across steps; attention evolves with reasoning (Wen et al., 2024; Li et al., 2023c), suggesting monitorable structure.
- **Metacognitive sensitivity.** Separating "hard to process" from "unknown" in uncertainty signals (Li et al., 2025b; Damani et al., 2025).

With those pieces, the present proxy story could seed predictors that trigger collaboration when load spikes—a step toward closed-loop load management.

**Developing universal proxies for cognitive load.**  Beyond runtime measurement, we need more **universal proxies** for design-time analysis. Our current measures can be influenced by model architecture and tokenization.  A critical direction is developing standardized "cognitive toolkits" that work reliably across diverse model families, potentially including gradient-based sensitivity analysis or direct elicitation methods as suggested by reviewers.

**Boundary conditions and task-adaptive collaboration.**  Our findings help delineate the **boundary conditions** under which CLT-based collaboration helps.  Benefits are most pronounced for high intrinsic cognitive load tasks.  Future work should develop principled methods to predict *a priori* which tasks require collaborative architectures, potentially through cognitive load estimation before execution.  This connects to the adaptive system vision above—determining not just *when* to collaborate during a task, but *which tasks* benefit from collaboration at all.

**Emergent collective dynamics.**  While *CoThinker* operationalizes mechanisms for collective cognition, the rich internal social dynamics remain underexplored.  Applying methods from **computational social science** to analyze interaction patterns—network evolution, transient leadership, consensus formation—could reveal deeper insights into artificial collective intelligence and potentially inform improved architectures.

**Human-AI collaboration.**  A particularly exciting direction is extending the framework to include human users as specialized agents, creating **human-AI cognitive systems** where the architecture actively manages cognitive load for both human and AI participants.  This could enable solving problems that neither could tackle alone, fostering true hybrid intelligence.

**Bidirectional benefits: LLMs as cognitive science research tools.**  Finally, we note the potential for **bidirectional knowledge transfer**.  While we apply cognitive science to improve LLMs, studying LLMs offers more controllable experimental paradigms than human studies.  Investigating how LLMs chunk infor-

mation, manage cognitive load, and develop collective intelligence could yield new insights about cognitive mechanisms that inform human cognitive science itself.

Addressing these future directions will advance our understanding of how to build truly collaborative and cognitively capable LLM-based systems while potentially contributing back to cognitive science through computational modeling.

