# OpenReview forum: "United Minds or Isolated Agents? Exploring Coordination of LLMs under Cognitive Load Theory"
_TMLR — Accepted by TMLR_

### Review · Reviewer_GzbH · 2026-04-08

**Summary Of Contributions:**

This paper argues that failures from heavy context engineering can be understood through the lens of Cognitive Load Theory (CLT): the model’s effective attention budget plays the role of working memory, and dense, high-interactivity prompts induce cognitive overload. On top of this framing, the paper introduces a pilot study using attention entropy and perplexity as load proxies, then proposes CoThinker, a multi-agent architecture with four main components: task-specific thinking-style assignment, a transactive memory system for compressed shared state, a communication moderator that sparsifies/reroutes peer messages, and a final synthesizer. Empirically, the paper evaluates the method on LiveBench, CommonGen-Hard, cross-model settings, and several ablations. The main strengths are the originality of the CLT framing, the unusually coherent mapping from theory to architecture, and the breadth of experiments and ablations. The main weaknesses are that the cognitive claims are still supported only indirectly, some performance claims are stronger than the evidence really warrants, and the paper does not yet make cost/fairness trade-offs as transparent as the accuracy results.

**Audience:**

Yes

**Audience Explanation:**

I think this paper would interest readers working on LLM agents, multi-agent coordination, context engineering, reasoning, and memory. The most valuable aspect is not just that the authors propose another agent framework, but that they try to ground specific design choices—specialization, summarized shared memory, and sparse communication—in an explicit theory. Even readers who are not fully persuaded by the CLT analogy may still find the architecture and ablations useful, especially because the paper also identifies a meaningful boundary condition: collaboration appears more helpful on high-load reasoning-style tasks than on straightforward instruction-following tasks.

**Broader Impact Concerns:**

I do not see a major unaddressed ethical issue, and the paper does include an ethics/limitations discussion.

**Claims And Evidence:**

Yes

**Claims Explanation:**

The paper provides several layers of evidence rather than relying on a single benchmark: a pilot study showing that attention entropy rises with task complexity and that instruction complexity helps harder tasks more than easy ones; main LiveBench results across three Gemini models; CommonGen-Hard results showing more stable multi-round behavior; cross-model experiments on additional model families; and targeted ablations for routing, style assignment, and TMS. Taken together, this is convincing evidence that the proposed architecture is useful for some high-load tasks, especially reasoning and data-analysis settings.

**Requested Changes:**

1. Temper the strongest causal/theoretical language. The current evidence shows that the results are consistent with a CLT interpretation and that CLT is a useful design lens. It does not yet establish that CLT is the unique or definitive explanation of context-rot behavior across LLMs. I would revise the framing accordingly, especially in the abstract, introduction, and conclusion.
2. Improve empirical transparency in the main results. Please report absolute task scores alongside normalized ones in the main text, and add confidence intervals or significance indicators for the key comparisons in Table 2. Appendix C includes bootstrap analysis, but the main presentation would be much easier to assess if uncertainty and effect sizes were visible where the main claims are made.
3. Improve empirical transparency in the main results. Please report absolute task scores alongside normalized ones in the main text, and add confidence intervals or significance indicators for the key comparisons in Table 2. Appendix C includes bootstrap analysis, but the main presentation would be much easier to assess if uncertainty and effect sizes were visible where the main claims are made.

---

> ### Author Response · Authors · 2026-04-29
>
> We thank the reviewer for the careful read and for highlighting *"the originality of the CLT framing, the unusually coherent mapping from theory to architecture, and the breadth of experiments and ablations."* The reviewer's specific request for absolute scores plus per-cell uncertainty in Table 2 is what drove the three-line cell layout (score / ratio / bootstrap SE) and the per-category Holm tests now reported in App. C.2 — both materially raise the evidentiary bar of the main result.
>
> **Weakness 1 + Recommendation 1** — *"the cognitive claims are still supported only indirectly"* / *"temper the strongest causal/theoretical language, especially in the abstract, introduction, and conclusion"*.
>
> We have softened the framing across abstract / Sec. 1 / Sec. 6. Two concrete excerpts:
> - *Abstract:* *"Drawing an analogy to CLT … We use CLT as a principled design lens … Our results are consistent with — not proof of — a CLT account."*
> - *Sec. 6:* *"treating the CLT-LLM correspondence as a functional analogy rather than a mechanistic claim."*
>
> We have re-framed the Instruction-Following losses as *evidence for* the predicted boundary, not as a flaw.
>
> **Weakness 2** — *"some performance claims are stronger than the evidence really warrants"*.
>
> We have softened Sec. 5 prose and restricted strong-significance language to the Holm-significant cells.
>
> **Weakness 3** — *"the paper does not yet make cost/fairness trade-offs as transparent as the accuracy results"*.
>
> We have added Sec. 5.7 + Table 6 (tokens within 0.89×–1.11× of MAD on every backbone). We have also documented in App. E.9 the apples-to-apples drop list (degenerate CoThinker traces; baseline empty/refusal outputs) applied uniformly across methods.
>
> **Recommendation 2 + Recommendation 3** — *"report absolute task scores alongside normalized ones, and add confidence intervals or significance indicators for the key comparisons in Table 2"*.
>
> We have rebuilt Table 2 so each cell shows three lines: absolute task score, per-subtask Flash-8B-IO-normalised ratio, and bootstrap SE; best per block in bold.
>
> We have used bootstrap SE rather than a sampling-CI because refinement uses greedy decoding, so seed variance is near zero; the reported SE therefore captures *question-sampling* variance — the relevant uncertainty for generalisation claims. We have noted that paired Holm-Bonferroni is the stricter end of the testing spectrum, and several near-significant cells do not pass at $\alpha{=}0.05$; we report what survives this stricter bar rather than the looser cell-wise intervals.
>
> We have reported per-category Holm tests in App. C.2 Table 13.

---

### Review · Reviewer_w53e · 2026-04-18

**Summary Of Contributions:**

This paper argues that many failures of heavily context-engineered LLM systems can be understood through the lens of Cognitive Load Theory (CLT). The authors propose that LLMs, like humans, have an effective working-memory bottleneck, and that performance degrades when context complexity exceeds that processing budget. To operationalize this idea, the paper introduces CoThinker, a multi-agent framework with three main design elements: task-adaptive thinking-style assignment, a transactive memory system that compresses shared state, and a communication moderator that limits and routes peer-to-peer information flow. The paper supports this framing with a pilot study using attention entropy and perplexity as cognitive-load proxies, then evaluates CoThinker against single-agent and multi-agent baselines on LiveBench and CommonGen-Hard, along with cross-model tests and component ablations. Overall, the paper’s main strength is that it offers a principled and fairly original design perspective for multi-agent coordination rather than another heuristic collaboration recipe. Its main weakness is that the central theoretical claim is still supported mostly indirectly: the CLT analogy is plausible and interesting, but the empirical evidence remains somewhat correlational, and some design choices appear more intuitively motivated than rigorously isolated.

**Additional Comments:**

I enjoyed reading this paper. I think its main contribution is not just the proposed architecture, but the attempt to introduce a more principled vocabulary for talking about why some multi-agent LLM systems help while others collapse under their own coordination burden. That is a worthwhile direction, and the paper contains several thoughtful design choices and useful experiments.

At the same time, I would encourage the authors to be careful not to overclaim what the current evidence establishes. The paper is strongest when it presents CLT as a productive and well-motivated design framework, and somewhat weaker when it implies that the theory has already been validated as the underlying mechanism of context failure in LLMs. With more cautious framing, clearer novelty positioning, and a bit more methodological sharpening, I think this could become a strong and memorable contribution.

**Audience:**

Yes

**Audience Explanation:**

Yes. I think this paper would be of interest to at least part of the TMLR audience because it sits at the intersection of LLM systems, multi-agent coordination, and theory-informed design. There is strong current interest in why multi-agent LLM systems help in some settings and fail in others, and this paper offers a unifying perspective that reframes those behaviors in terms of intrinsic and extraneous load rather than only prompting strategy or coordination efficiency. Even readers who are not fully persuaded by the CLT analogy may still find the design principles behind CoThinker useful.

I also think the paper is relevant because it goes beyond proposing yet another agent framework and instead asks a more general question: how should collaborative LLM systems be designed when attention and context are scarce resources? That is a meaningful systems question for the community. The paper is also broad enough in evaluation to make the discussion interesting to researchers working on prompting, agent communication, memory mechanisms, and benchmark-driven evaluation.

**Broader Impact Concerns:**

I do not have major unaddressed broader-impact concerns beyond the standard risks associated with stronger multi-agent generation systems. The paper already notes that better coordinated agent systems could be used to scale misleading or low-quality content generation, intensify automation pressure in knowledge work, or concentrate capability among well-resourced actors. I think that discussion is reasonable and proportionate for this type of methods paper.

My main suggestion here is not that the broader-impact section is missing, but that it could be made slightly more concrete. In particular, the authors could briefly discuss whether load-managed multi-agent systems might make it easier to automate persuasive or strategically adaptive content generation at scale, and whether certain deployment settings would warrant human oversight, rate limits, or domain-specific safeguards. That would strengthen the ethics discussion without requiring a major expansion.

**Claims And Evidence:**

Yes

**Claims Explanation:**

The paper provides a meaningful amount of evidence for its main empirical claim that the proposed coordination strategy can outperform several standard baselines on complex tasks. In particular, the LiveBench results show gains over IO, CoT, Self-Refine, MAD, and DMAD in several higher-load categories, and the CommonGen-Hard results plus the round-by-round analysis are consistent with the authors’ claim that structured coordination can avoid the degradation seen in less controlled multi-agent interaction. The cross-model results and the communication-moderator ablations also strengthen the empirical case that the method is not purely anecdotal.

That said, I still view the evidence as supportive rather than fully definitive for the broader theoretical claim. The paper’s strongest framing is that CLT explains context overload in LLM systems, but the evidence for that interpretation relies on proxy variables such as attention entropy and perplexity, which the paper itself acknowledges are mainly diagnostic and not direct runtime measures of cognitive state. Similarly, some architectural choices—such as the thinking-style mechanism, the TMS summary structure, and the small-world communication framing—are sensible and well motivated, but it remains somewhat difficult to disentangle which gains come from the CLT grounding specifically versus from generally good coordination and summarization design. So I believe the claims are sufficiently supported for publication discussion, but several of the higher-level explanatory claims should be stated a bit more cautiously.

**Requested Changes:**

**Critical**
1. Clarify the boundary between a useful explanatory theory and a proven causal account. The paper currently presents CLT as a strong interpretive foundation for LLM coordination, but much of the evidence is proxy-based and correlational. I recommend softening any wording that suggests the paper has established CLT as the definitive causal explanation for context rot, and instead frame it as a well-supported and productive design lens.

2. Strengthen the discussion of what exactly is novel beyond prior multi-agent coordination techniques. The current architecture combines specialization, summarization/shared memory, and moderated communication, all of which are individually familiar ideas. The paper should more explicitly isolate what is uniquely enabled by the CLT framing and why that framing leads to better design decisions than prior heuristic systems.

3. Expand methodological clarity around the pilot study and proxies. The attention-entropy and perplexity results are interesting, but the paper should be more explicit in the main text about their limitations, what they do and do not measure, and why they should be interpreted as evidence for cognitive load rather than more generic indicators of task difficulty or uncertainty.
Important but not strictly critical: Improve the presentation of task-dependent failures. The paper does mention that CoThinker underperforms on instruction-following tasks because coordination overhead can dominate when intrinsic load is low. This is a valuable negative result and deserves a more prominent and sharper discussion, since it helps define when the method should and should not be used.

**Important but not critical:**

1. Provide a more detailed cost/latency analysis. Since CoThinker is a multi-agent framework with multiple rounds, agents, memory updates, and communication moderation, readers would benefit from a clearer understanding of inference-time overhead relative to the performance gains.

2. Strengthen the ablation narrative. The current ablations are helpful, especially for the communication moderator, thinking styles, and TMS, but the paper would benefit from a clearer causal interpretation of why some components help more on some model families than others, especially where newer GPT-style models resist intermediate reasoning traces.

**Strengthening suggestion:**

1. Add a short section connecting the design principles to practical deployment guidance. For example, the paper could summarize simple heuristics for when single-agent prompting is sufficient, when collaborative architectures are worthwhile, and when communication sparsity matters most.

2. Tighten some claims around “working memory” analogies. The analogy is thought-provoking, but a bit of precision in wording would help avoid overstating equivalence between human cognitive constructs and transformer attention behavior.

---

> ### Author Response · Authors · 2026-04-29
>
> We thank the reviewer for the generous read and especially for recognizing that the paper's main contribution is *"the attempt to introduce a more principled vocabulary for talking about why some multi-agent LLM systems help while others collapse under their own coordination burden."* We thank the reviewer's suggestion of framing CLT as a *"productive and well-motivated design framework"* which shaped the abstract, Sec. 1, and Sec. 6.
>
> **Weakness 1 + Concern 1** — *"the empirical evidence remains somewhat correlational"* / *"soften wording that suggests CLT is the definitive causal explanation; frame it as a productive design lens"*.
>
> We have softened the framing across abstract / Sec. 1 / Sec. 6. Two concrete excerpts:
> - *Abstract:* *"Drawing an analogy to CLT … We use CLT as a principled design lens … Our results are consistent with — not proof of — a CLT account."*
> - *Sec. 6:* *"treating the CLT-LLM correspondence as a functional analogy rather than a mechanistic claim."*
>
> We have re-framed the Instruction-Following losses as *evidence for* the predicted boundary, not as a flaw.
>
> **Weakness 2** — *"higher-level explanatory claims should be stated more cautiously"*.
>
> We have done a rewrite of some verbs across Sec. 5 (`demonstrates` → `shows`; `confirming` → `consistent with`; `validate` → `support`).
>
> **Concern 2** — *"strengthen the discussion of what is novel beyond prior multi-agent coordination techniques"*.
>
> We have added at the end of Sec. 4: *"three choices follow directly from the CLT lens: thinking styles are generated adaptively per task rather than fixed as personas, peer input is routed through a moderator that caps integration per round, and the system is expected to help only where intrinsic load is high enough to justify coordination cost — a task-dependent boundary the 'more agents are better' view does not predict."*
>
> **Concern 3** — *"be more explicit in the main text about [the proxies'] limitations"*.
>
> We have added in Sec. 3.2: *"Both proxies are diagnostic rather than runtime signals — perplexity requires ground-truth answers and attention entropy is model-internal — and we use them as supporting evidence for the analogy rather than as operational controls."*
>
> **Concern 4** — *"improve the presentation of task-dependent failures"*.
>
> We have stated the Instruction-Following losses directly in the Sec. 5.2 main results paragraph and framed them as confirming the predicted boundary.
>
> **Improvement 1** — *"provide a more detailed cost/latency analysis"*.
>
> We have added Sec. 5.7 + Table 6 (tokens within 0.89×–1.11× of MAD on every backbone).
>
> **Improvement 2** — *"strengthen the ablation narrative; clearer causal interpretation of why some components help more on some model families"*.
>
> We have added in Sec. 5.6: *"backbones that suppress intermediate reasoning traces (common in newer safety-tuned models) degrade TMS effectiveness; the right fix is to design TMS prompts that synthesise from agents' final-answer content rather than ask the backbone to expose its reasoning, since the latter triggers the suppression behaviour."* We attribute the cross-family variation to safety-RL post-training in newer GPT models suppressing the intermediate traces TMS was originally written to consume, and we point to a prompt-side adjustment as the actionable response.
>
> **Suggestion 1** — *"add deployment guidance heuristics"*.
>
> We have added at the end of Sec. 6: *"(i) single-agent CoT suffices when intrinsic load is low; (ii) multi-agent coordination pays off when the task decomposes into distinct reasoning styles or sub-problems; (iii) the moderator matters most when raw peer output would otherwise dominate each agent's context."*
>
> **Suggestion 2** — *"tighten claims around 'working memory' analogies"*.
>
> We have rephrased Sec. 3.1 as *"structural parallels we treat as a design analogy rather than a mechanistic claim about transformer internals."*
>
> **Broader Impact** — *"discuss whether load-managed multi-agent systems might lower the cost of persuasive content at scale"*.
>
> We have added in App. F.1 a concrete misuse mechanism: load-managed multi-agent systems are explicitly designed to compose information across many sources without exceeding any single agent's attention budget, and the same capability can be turned toward constructing tailored persuasive content that integrates a target audience's priors at once — with different agents specialising in factual framing, emotional appeal, and social proof.

---

> > ### Comment · Reviewer_w53e · 2026-04-29
> > **Thanks for the clarification and Updates**
> >
> > Your responses have been helpful, and at this stage I do not have further questions. I appreciate the clarifications you have provided and will consider them in my final assessment.

---

### Review · Reviewer_hpZ3 · 2026-04-23

**Summary Of Contributions:**

‌This paper presents a Cognitive Load Theory (CLT) framework for LLM limitations. It introduces the CoThinker architecture based on the CLT framework, and provides empirical evidence demonstrating CoThinker's effectiveness across LLMs (including Gemini, GPT, Qwen families) and challenging benchmarks (LiveBench, CommonGen-Hard).

The key strength lies in that it grounds the architectural design in the well-established principles of CLT from cognitive science. This moves beyond heuristic-driven multi-agent systems to a ‌principled, cognitively-informed approach‌.

Potential Limitations / Drawbacks:
1. The cognitive load proxies (attention entropy, perplexity) are ‌diagnostic and not operational‌ for real-time system control. They can be influenced by confounds and require ground-truth for perplexity calculation, limiting their direct use in guiding the system during inference.
2. As with most multi-agent systems, CoThinker introduces significant ‌computational cost‌ (more API calls, longer generation times, increased token usage) compared to single-agent prompting.
3. The effectiveness of certain components (notably the Transactive Memory System - TMS) is noted to vary across different LLM models (e.g., less effective for newer GPT models), indicating the architecture might require ‌model-specific tuning or prompting strategies‌.

**Audience:**

Yes

**Audience Explanation:**

This paper shall likely interest researcher from both AI and cognitive science communities.

**Claims And Evidence:**

Yes

**Claims Explanation:**

The pilot study validates the problem diagnosis (cognitive overload), which justifies the solution design (CoThinker), and the main experiments + ablations confirm that the solution works for the reasons the theory predicts. This tight, evidence-backed loop is what makes the paper's arguments compelling.

**Requested Changes:**

1. Quantify the increased inference time, token usage, and API calls compared to baselines. Discuss how this cost might be amortized for critical tasks vs. being a barrier for low-latency applications.
2. Discuss the model-dependence of the TMS (especially for models that refuse intermediate reasoning) as a practical deployment challenge.
3. More explicitly define the "cognitive load profile" of tasks where CoThinker excels (high intrinsic load, reasoning-heavy) vs. where it may not (low intrinsic, execution-focused).
4. Acknowledge more explicitly that perplexity requires ground truth and attention entropy is model-internal and not a real-time control signal. Frame the search for operational cognitive load metrics as a major future direction.

---

> ### Author Response · Authors · 2026-04-29
>
> We thank the reviewer for the careful read and for recognizing that the work *"grounds the architectural design in the well-established principles of CLT … beyond heuristic-driven multi-agent systems to a principled, cognitively-informed approach."* The reviewer's push to quantify cost and to make the cognitive-load profile of tasks explicit drove two of the most concrete additions in this revision (Sec. 5.7 Table 6 and the elaboration in Sec. 5.2).
>
> **Limitation 1 + Recommendation 4** — *"the cognitive load proxies … are diagnostic and not operational"* / *"frame the search for operational cognitive load metrics as a major future direction"*.
>
> We have added in Sec. 3.2: *"Both proxies are diagnostic rather than runtime signals — perplexity requires ground-truth answers and attention entropy is model-internal — and we use them as supporting evidence for the analogy rather than as operational controls."* We have clarified that CoThinker's runtime control (moderator routing, TMS compression) is rule-based and does not depend on either proxy. We have also elevated the search for inference-time CL metrics to a major future direction in App. F.2.
>
> **Limitation 2 + Recommendation 1** — *"CoThinker introduces significant computational cost"* / *"quantify the increased inference time, token usage, and API calls; discuss amortization vs. low-latency barrier"*.
>
> We have added Sec. 5.7 *Cost and Latency* with Table 6. We have shown that CoThinker uses tokens within 0.89×–1.11× of MAD on every backbone — comparable to standard multi-agent baselines. We have also clarified that CoThinker runs its $M$ agents in parallel within each round, so wall-clock latency is dominated by round count, not agent count. The total tokens stay close to MAD because the moderator caps how much each agent must integrate per round. We have added the deployment context: *"the multi-agent overhead is worth paying on reasoning-heavy queries, where the accuracy gain is largest, but not on low-intrinsic-load tasks, so the cognitive-load profile (Sec. 5.2) doubles as a deployment-time routing signal."*
>
> **Limitation 3 + Recommendation 2** — *"the effectiveness of certain components (notably TMS) … vary across different LLM models"* / *"discuss the model-dependence of the TMS … as a practical deployment challenge"*.
>
> We have added to Sec. 5.6: *"backbones that suppress intermediate reasoning traces (common in newer safety-tuned models) degrade TMS effectiveness; the right fix is to design TMS prompts that synthesise from agents' final-answer content rather than ask the backbone to expose its reasoning, since the latter triggers the suppression behaviour."* Framing this as a deployment-time prompt-engineering decision (rather than a per-model retraining requirement) gives practitioners a concrete adjustment path on safety-tuned backbones.
>
> **Recommendation 3** — *"more explicitly define the 'cognitive load profile' of tasks where CoThinker excels vs. where it may not"*.
>
> We have added an explicit elaboration in Sec. 5.2: *"the cognitive-load profile sorts a task by how much working-memory pressure it imposes before any coordination overhead: high-intrinsic-load tasks are reasoning-heavy and decomposable into distinct sub-problems (Math, Data Analysis, Reasoning, Language), while low-intrinsic-load tasks are execution-focused and rubric-following (Instruction-Following), so a single agent already handles them well."* We have validated this split with per-category Holm-corrected paired-bootstrap tests (App. C.2 Table 13): **16 of 25 comparisons are Holm-significant**, and **all 3 Holm losses fall on Instruction-Following** — the predicted low-intrinsic-load boundary.

---

### Decision · Action_Editor_veCQ · 2026-05-31

**Recommendation:** Accept as is

**Additional Comments:**

This paper proposes a new multi-agent LLM design motivated by cognitive load theory. The key idea is that similarly to humans, a single LLM agent gets easily overwhelmed by a large amount of unstructured information put in context when solving complex problems. To solve a complex problem, a better idea is to have task-specific agents that communicate and solve it jointly. Cognitive load theory suggests that the memory of the agents should be compressed into a state and that the communication of the agents should be orchestrated, to reduce the cognitive load.

Reviewers already liked the first version of the paper. One reason is that the proposed design is less heuristic than other multi-agent papers. This topic will also grow in importance as more frontier LLMs become closed. All three reviewers, perhaps surprisingly, had the following comment:

* **Gap between cognitive load theory and experiments:** All experiments are on proxies to cognitive load: attention entropy and perplexity. Therefore, they do not establish that the current design, based on cognitive load theory, minimizes the cognitive load. The reviewers suggested that the authors weaken their claims. The authors weakened their claims in the revised paper.

The reviewers had other suggestions, such as reporting latency and improving discussions. The authors revised the paper to take these into account.

This is a good paper on a timely topic. All reviewers lean towards acceptance. I suggest accepting this paper.

**Audience:**

Yes

**Audience Explanation:**

LLM agents are a popular topic. Multi-agent systems are one way of bringing structure and modularity to the design. As more frontier LLMs become closed, this topic will again in importance.

**Claims And Evidence:**

Yes

**Claims Explanation:**

This paper proposes a new multi-agent LLM design motivated by cognitive load theory. The approach is evaluated on two benchmarks (LiveBench and CommonGen-Hard), three model families (Gemini, Qwen, and GPT), and compared to multiple baselines.